# Hyphal Growth in *Trichosporon asahii* Is Accelerated by the Addition of Magnesium

Keita Aoki,[a] Kosuke Yamamoto,[d] Moriya Ohkuma,[b] Takashi Sugita,[c] Naoto Tanaka,[d] Masako Takashima[a]

[a]Laboratory of Yeast Systematics, Tokyo NODAI Research Institute, Tokyo University of Agriculture, Setagaya, Tokyo, Japan
[b]Japan Collection of Microorganisms, RIKEN BioResource Research Center, Tsukuba, Ibaraki, Japan
[c]Department of Microbiology, Meiji Pharmaceutical University, Kiyose, Tokyo, Japan
[d]Department of Molecular Microbiology, Faculty of Life Sciences, Tokyo University of Agriculture, Setagaya, Tokyo, Japan

**ABSTRACT** Fungal dimorphism involves two morphologies: a unicellular yeast cell and a multicellular hyphal form. Invasion of hyphae into human cells causes severe opportunistic infections. The transition between yeast and hyphal forms is associated with the virulence of fungi; however, the mechanism is poorly understood. Therefore, we aimed to identify factors that induce hyphal growth of *Trichosporon asahii*, a dimorphic basidiomycete that causes trichosporonosis. *T. asahii* showed poor growth and formed small cells containing large lipid droplets and fragmented mitochondria when cultivated for 16 h in a nutrient-deficient liquid medium. However, these phenotypes were suppressed via the addition of yeast nitrogen base. When *T. asahii* cells were cultivated in the presence of different compounds present in the yeast nitrogen base, we found that magnesium sulfate was a key factor for inducing cell elongation, and its addition dramatically restored hyphal growth in *T. asahii*. In *T. asahii* hyphae, vacuoles were enlarged, the size of lipid droplets was decreased, and mitochondria were distributed throughout the cell cytoplasm and adjacent to the cell walls. Additionally, hyphal growth was disrupted due to treatment with an actin inhibitor. The actin inhibitor latrunculin A disrupted the mitochondrial distribution even in hyphal cells. Furthermore, magnesium sulfate treatment accelerated hyphal growth in *T. asahii* for 72 h when the cells were cultivated in a nutrient-deficient liquid medium. Collectively, our results suggest that an increase in magnesium levels triggers the transition from the yeast to hyphal form in *T. asahii*. These findings will support studies on the pathogenesis of fungi and aid in developing treatments.

**IMPORTANCE** Understanding the mechanism underlying fungal dimorphism is crucial to discern its invasion into human cells. Invasion is caused by the hyphal form rather than the yeast form; therefore, it is important to understand the mechanism of transition from the yeast to hyphal form. To study the transition mechanism, we utilized *Trichosporon asahii*, a dimorphic basidiomycete that causes severe trichosporonosis since there are fewer studies on *T. asahii* than on ascomycetes. This study suggests that an increase in $Mg^{2+}$, the most abundant mineral in living cells, triggers growth of filamentous hyphae and increases the distribution of mitochondria throughout the cell cytoplasm and adjacent to the cell walls in *T. asahii*. Understanding the mechanism of hyphal growth triggered by $Mg^{2+}$ increase will provide a model system to explore fungal pathogenicity in the future.

**KEYWORDS** magnesium, *Trichosporon asahii*, hyphal growth, fungal dimorphism

Address correspondence to Masako Takashima, mt207623@nodai.ac.jp, or Keita Aoki, ka207755@nodai.ac.jp.

The authors declare no conflict of interest.

Fungi exhibit various cellular morphologies to adapt to different natural habitats (1). The most common phenomenon of adaptation involves dimorphism. In dimorphism, fungi exhibit two types of morphology: a unicellular yeast cell and a multicellular filamentous hypha (2). Since hyphal cells invade the epithelial cells of human organs

and cause several opportunistic infections in immunocompromised patients (3), the transition mechanism between yeast and hyphal forms is of great interest in studies on Ascomycota and Basidiomycota. Yeasts transition into hyphae in response to various external stimuli. In *Candida albicans*, a representative dimorphic ascomycete, various environmental stimuli induce hypha-associated genes via several signal transduction pathways and induce asexual hypha formation (1, 2, 4). In *Schizosaccharomyces japonicus*, a nonpathogenic fission yeast, asexual hyphae are formed upon DNA damage stress (5) and exposure to fruit extracts (6). In contrast, *Cryptococcus neoformans*, a human-pathogenic basidiomycete, forms true hyphae and basidia at the tip of the hyphae by sensing pheromones during the mating process (7, 8). Further, *C. neoformans* shows pseudohyphal growth, when cultured with amoeba (9), as a survival strategy to escape the host defense system (10). The pseudohyphal growth is regulated by the cyclic AMP signaling pathway (11). *Trichosporon asahii* is a dimorphic basidiomycete that occasionally causes trichosporonosis in immunocompromised patients (12–17). *T. asahii* is widely distributed in soil, leaf mold, and decayed wood (12). In these environments, *T. asahii* cells exhibit distinct morphological forms that include yeast, hyphal, and arthroconidial forms (15). The presence of *T. asahii* hyphae in blood vessels has been associated with infection (18); arthroconidia contribute to biofilm formation by promoting cell adhesion (19). However, the mechanism by which different forms of *T. asahii* transition into one another is not fully understood.

Cations are involved in several cellular processes. They function as cofactors for various enzymes and DNA-binding proteins. Particularly, zinc-binding proteins represent one of the largest families of transcription factors in eukaryotic cells. In fungi, zinc-binding transcription factors regulate filament formation (20, 21). In *T. asahii*, the chelation of zinc by *N,N,N′,N′*-tetrakis (2-pyridylmethyl) ethylenediamine inhibits biofilm and hyphal formation (22). In contrast, *C. albicans* hyphal formation is enhanced by the addition of $Mg^{2+}$ rather than $Zn^{2+}$ (23). $Mg^{2+}$ metabolism is related to gene expression because magnesium exists *in vitro*, in the sirtuin-nucleosome complex, a histone deacetylase (24). Sirtuin 2 is involved in hyphal growth of *C. albicans* (25). Therefore, hyphal formation may be induced by magnesium; however, the mechanism is not fully understood.

Hyphal growth in pathogenic fungi requires functional mitochondria (26). Mitochondria undergo fission and fusion constantly and dynamically (27). Mitochondrial motility depends on cytoskeletal networks. Actin filaments drive mitochondrial segregation in *Saccharomyces cerevisiae* (28, 29) and humans (30). In contrast, the mitochondrial positioning in *Schizosaccharomyces pombe* is driven by the interaction with microtubules and spindle poles (31). These cytoskeletal networks facilitate the transfer of mitochondria toward the growing tip of hyphae. Mitochondria are present at a high density in the growing tip of the ascomycete *Neurospora crassa* (32). The polarized distribution of mitochondria is also detected in the pollen tube and neurons. In pollen tubes, numerous mitochondria are enriched in the subapical region behind the vesicle-filled tip (33, 34). In neuronal cells, the establishment of polarity is closely related to mitochondrial distribution (35). Moreover, it is presumed that mitochondria, which are distributed at a neuronal growth tip, may function as an energy source (36, 37). However, the mitochondrial phenotypes in *T. asahii* still remain to be explored.

Mitochondria are organelles packed in a lipid bilayer in most eukaryotic cells and perform aerobic respiration to generate ATP (38). Mitochondrial function requires acetyl coenzyme A (acetyl-CoA) supplied via interaction between mitochondria and lipid droplets (39). Acetyl-CoA is required in the citric acid cycle to produce NADH that is utilized for the electron transport chain to produce ATP (40). When cultivated in a medium containing nonfermentable carbon substrates (41), *S. cerevisiae* cells show a growth defect exhibiting a "petite" phenotype in which ATP synthesis is compromised (42). In the "petite" phenotype produced by synthetic mutations in *OM45* and mitochondrial Rho GTPase 1 in *S. cerevisiae*, mitochondrial morphology is defective and fragmented (43). Therefore, mitochondrial respiration and structure are linked, indicating that a fragmented structure of mitochondria leads to malfunction in respiration in

*S. cerevisiae*. In addition, a decrease in $Mg^{2+}$ influx causes malfunction in mitochondria. The "petite" phenotype is also caused by disruption in genes *MRS2* and *LPE10*, of the CorA family, which encode magnesium transporters embedded in the inner mitochondrial membrane (44, 45). A pioneer work using *S. pombe* also indicated that mitochondria require magnesium to maintain adequate structure and respiration since the small and round structure and decreased respiration are produced due to depletion of magnesium from the medium (46).

Since the mechanism of yeast-to-hypha transition in fungi remains poorly understood, we aimed to identify factors that induce this transition in *T. asahii* and investigated their effects on cellular components in this study. Our results indicate that an increase in magnesium levels triggers the transition of yeast to hypha and mitochondrial distribution in *T. asahii*.

## RESULTS

***T. asahii* shows slow growth and produces large lipid droplets in Sabouraud medium.** To study the cell morphological phenotype of *T. asahii*, we examined the cell length of *T. asahii* JCM 2466 in two liquid media: Sabouraud medium and yeast extract-peptone-dextrose (YPD) medium. JCM 2466 cells were cultivated for 16 h in each medium and fixed with 70% ethanol before observation under the microscope. The length of JCM 2466 cells cultivated in Sabouraud medium was 11.05 $\pm$ 11.19 $\mu$m ($n = 406$), whereas that of JCM 2466 cells cultivated in YPD medium was 23.78 $\pm$ 14.04 $\mu$m ($n = 411$) (Fig. 1A). This suggests that hyphal growth was inhibited in Sabouraud medium. Additionally, JCM 2466 cells cultivated in Sabouraud medium produced droplets that were stained by Nile red and not by 4′,6-diamidino-2-phenylindole (DAPI) (Fig. 1A and B). The phenotype was not prominent in cells cultivated in YPD (Fig. 1A), and no signals were detected in unstained cells (Fig. 1A). Therefore, JCM 2466 cells accumulated lipid droplets in Sabouraud medium, but not in YPD medium. The lipid droplets were also detected as a protruded round structure in the differential interference contrast (DIC) image, and the structure was useful as a marker of lipid droplets (Fig. 1B).

Next, we examined the growth rates of JCM 2466 cells in Sabouraud medium by recording the optical density values at 660 nm ($OD_{660}$) continuously for 3 days. The growth rate of cells cultivated in Sabouraud medium was considerably lower than that of cells cultivated in YPD medium (Fig. 1C). After culture for 72 h, the $OD_{660}$ values of Sabouraud and YPD media were 0.422 and 2.072, respectively (Fig. 1C), suggesting that JCM 2466 cells cultivated in Sabouraud medium showed poor growth compared with cells cultivated in YPD medium. The delay in growth of JCM 2466 cells in Sabouraud medium was suppressed by adding yeast nitrogen base (YNB) (Fig. 1C), and the length of JCM 2466 cells increased significantly upon the addition of YNB ($P < 10^{-5}$) (Fig. 2A; see the supplemental table in the supplemental material). The results suggested that YNB contained a key factor for inducing hyphal growth in *T. asahii*.

**Magnesium supplementation induces hyphal growth in *T. asahii*.** To isolate the key factor required for hyphal growth of *T. asahii* from the YNB, we first classified YNB components into four groups: amino acids, trace elements, salts, and vitamins. Subsequently, we determined the effect of supplementation of each group on hyphal growth compared to that observed in cells cultured in the medium supplemented with YNB. We cultivated JCM 2466 cells for 16 h in Sabouraud medium containing compounds from each group and measured the lengths of live cells under the microscope. Cells supplemented with the salt group were significantly longer than those supplemented with other groups ($P < 10^{-5}$) or cultured in Sabouraud medium alone ($P < 10^{-5}$) (supplemental table). The numbers of cells were classified into 11 groups of cell length with the following ranges: 0 to 10 $\mu$m, 10 to 20 $\mu$m, 20 to 30 $\mu$m, 30 to 40 $\mu$m, 40 to 50 $\mu$m, 50 to 60 $\mu$m, 60 to 70 $\mu$m, 70 to 80 $\mu$m, 80 to 90 $\mu$m, 90 to 100 $\mu$m, and longer than 100 $\mu$m. In the cell length classification, there was a reduction in the number of cells with a length of $<20$ $\mu$m in response to the addition of the salt group (Fig. 2A; supplemental table). Conversely, the cell lengths did not increase when the salt group alone was excluded from the YNB preparation ($P = 0.26$) (supplemental table). In the cell length classification, the number of cells with a

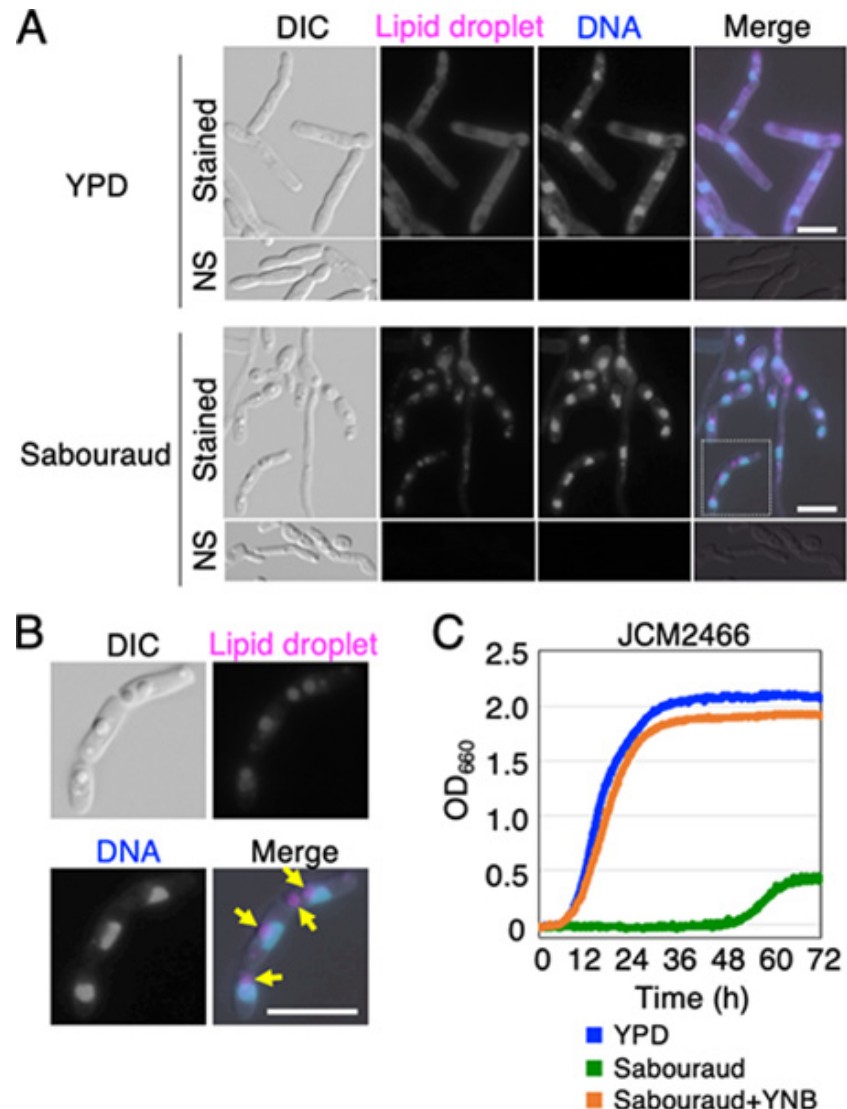

**FIG 1** *T. asahii* formed small cells containing large lipid droplets and showed slow growth in Sabouraud medium. (A) Lipid droplets and DNA of JCM 2466 cells cultivated in YPD and Sabouraud media for 16 h at 25°C were stained with Nile red and DAPI, respectively. NS, not-stained control. (B) The cell surrounded by a dotted line in panel A is enlarged. Lipid droplets, which are not stained by DAPI, are indicated by arrows. Scale bar is 10 $\mu$m. (C) Growth rates of JCM 2466 cells cultivated in three liquid media, YPD medium (blue line), Sabouraud medium (green line), and Sabouraud medium containing YNB (orange line), were measured every 10 min using a bio-photorecorder to determine the $OD_{660}$ for 72 h at 25°C. DIC, differential interference contrast microscopy; YPD, yeast extract-peptone-dextrose medium; YNB, yeast nitrogen base; $OD_{660}$, optical density at 660 nm.

length of <20 $\mu$m increased with the exclusion of the salt group (Fig. 2A; supplemental table). Furthermore, the lipid droplets in JCM 2466 cells cultured in Sabouraud medium diminished in size when the salt group was added but not when other groups were added (see Fig. S1 in the supplemental material). These results indicated that the salt group contained a key factor inducing hyphal growth of *T. asahii* in Sabouraud medium.

To identify the key factor, we screened the compound in the salt group that induced the hyphal growth of JCM 2466 cells. Dropout experiments of the salt group were performed as described for the four groups. The cell lengths were significantly shortened when $MgSO_4$ alone was excluded from the salt group ($P < 10^{-5}$) (supplemental table). The cells with a length of <20 $\mu$m increased in the cell length classification (Fig. 2B; supplemental table). Cells cultured in Sabouraud medium supplemented with 4.15 mM $MgSO_4$ (defined as Sabouraud+Mg medium) were significantly longer

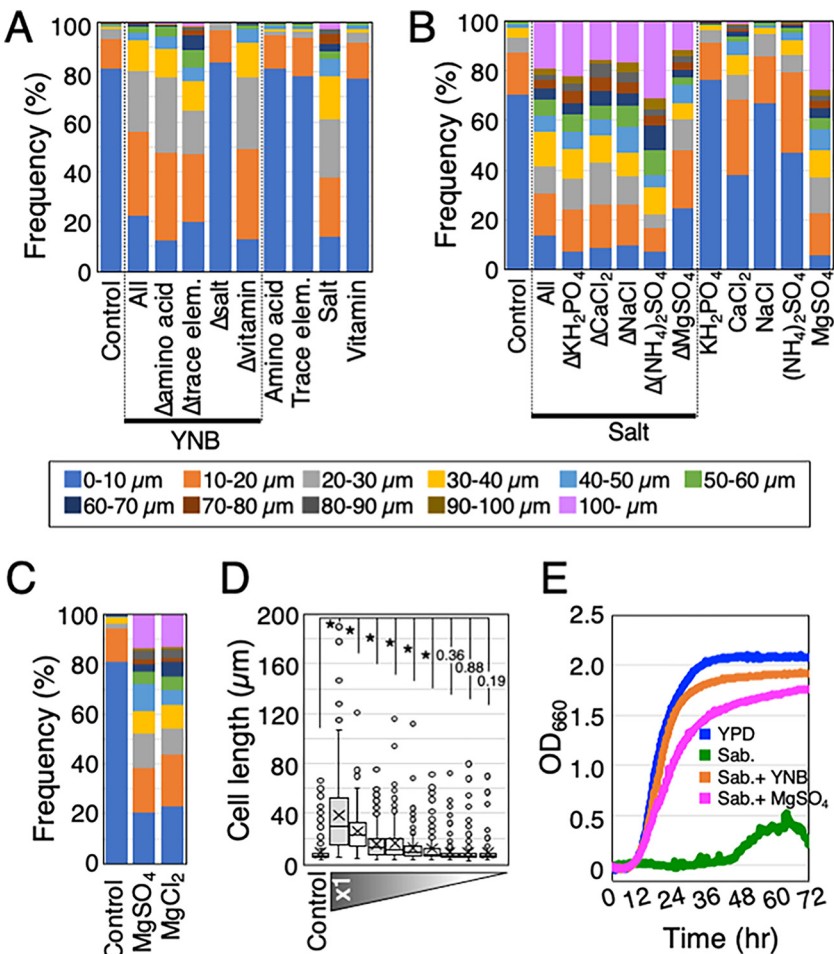

**FIG 2** Magnesium sulfate treatment induced hyphal growth in *T. asahii*. (A) The length of JCM 2466 cells was measured after cultivation in the control, Sabouraud medium containing each ingredient (amino acids, trace elements, salts, or vitamins) of YNB, and Sabouraud medium containing each dropout of the ingredient and then shown as the cell length classification. The amino acids were L-histidine monohydrochloride, LD-methionine, and LD-tryptophan. The trace elements were boric acid, manganese sulfate, zinc sulfate, ferric chloride, sodium molybdate, potassium iodide, and copper sulfate. The salts were ammonium sulfate, monopotassium phosphate, magnesium sulfate, sodium chloride, and calcium chloride. The vitamins were inositol, niacin, pyridoxine, thiamine, calcium pantothenate, riboflavin, *p*-aminobenzoic acid, folic acid, and biotin. (B) Lengths of JCM 2466 cells were measured after cultivation in control, Sabouraud medium containing each ingredient [$KH_2PO_4$, $CaCl_2$, NaCl, $(NH_4)_2SO_4$, or $MgSO_4$] of the salt group, and Sabouraud medium containing a dropout of each ingredient of the salt group and then shown as the cell length classification. Δ indicates dropout of the corresponding ingredient. (C) Lengths of JCM 2466 cells cultivated in the control medium, Sabouraud medium containing 4.15 mM $MgSO_4$, and Sabouraud medium containing $MgCl_2$ were measured and shown as the cell length classification. (D) Lengths of JCM 2466 cells were measured after cultivation in Sabouraud medium containing $MgSO_4$ at different concentrations for 16 h at 25℃. The concentration of $MgSO_4$ in the ×1 column was 4.15 mM. The dilution series of $MgSO_4$ was 1/10 (415 nM), 1/100 (41.5 nM), 1/200 (20.75 nM), 1/400 (10.38 nM), 1/600 (6.92 nM), 1/800 (5.19 nM), 1/1,000 (4.15 nM), and 1/10,000 (415 pM). Statistical differences between samples were calculated using the Mann-Whitney U test. The P values indicated by asterisks are significant at $P < 0.05$. The cross marks indicate each average length. (E) Growth rates of JCM 2466 cells cultivated in four liquid media, including YPD (blue line), Sabouraud medium (green line), Sabouraud medium containing YNB (orange line), and Sabouraud medium containing 4.15 mM $MgSO_4$ (pink line), were measured every 10 min using the bio-photorecorder for 72 h at 25℃. Control, Sabouraud medium without supplement; YPD, yeast extract-peptone-dextrose medium; Sab., Sabouraud medium; YNB, yeast nitrogen base.

than those grown in medium supplemented with $KH_2PO_4$, $CaCl_2$, NaCl, $(NH_4)_2SO_4$, or the control ($P < 10^{-5}$) (supplemental table). In response to the addition of 4.15 mM $MgSO_4$, there was a reduction in the number of cells with a length of <20 $\mu$m in the cell length classification, whereas the number of the cells with a length of ≥100 $\mu$m

increased (Fig. 2B; supplemental table). Furthermore, the lipid droplets of the cells cultured in Sabouraud medium diminished in size upon addition of 4.15 mM $MgSO_4$ (Fig. S2). Similarly, cells cultured in Sabouraud medium elongated significantly upon addition of 4.15 mM $MgCl_2$ ($P < 10^{-5}$) (supplemental table). The increases in cell lengths upon addition of 4.15 mM $MgSO_4$ and 4.15 mM $MgCl_2$ were comparable ($P = 0.61$), and the cell length classifications were similar (Fig. 2C; supplemental table). The size of the lipid droplets decreased in JCM 2466 cells cultured in Sabouraud medium supplemented with 4.15 mM $MgCl_2$ (Fig. S3). Therefore, magnesium was the key factor inducing hyphal growth of *T. asahii* in Sabouraud medium.

We determined whether the restoration of hyphal growth via addition of $MgSO_4$ depended on its concentration. We prepared a dilution series of $MgSO_4$ in Sabouraud medium and cultured JCM 2466 cells for 16 h at 25°C in Sabouraud medium supplemented with the dilution series of $MgSO_4$ and then measured the cell lengths. The cell length gradually shortened as $MgSO_4$ concentration decreased and was significantly elongated when 6.92 $\mu$M $MgSO_4$ was added to Sabouraud medium ($P < 10^{-5}$) (Fig. 2D; supplemental table). Therefore, the lowest concentration of $MgSO_4$ required to induce the elongation of JCM 2466 cells was 6.92 $\mu$M when the cells were cultivated for 16 h in Sabouraud medium.

To examine whether the growth delay of JCM 2466 cells in Sabouraud medium is restored by addition of $MgSO_4$, a growth curve was generated for 72 h at 25°C using a bio-photorecorder. The low growth rate of JCM 2466 cells in Sabouraud medium was dramatically increased by the addition of 4.15 mM $MgSO_4$. The $OD_{660}$ value of JCM 2466 cells was 1.75 after continuous culture for 72 h in Sabouraud+Mg medium, higher than that of cells grown in Sabouraud medium (0.229) (Fig. 2E). The value was comparable for cells grown in YPD medium (2.072) and for those grown in Sabouraud medium supplemented with YNB (1.914) (Fig. 2E). In addition, a colony formation assay was performed to examine the viability of cells cultivated in YPD, Sabouraud, and Sabouraud+Mg media. Whereas we detected 100% viability for the cells cultivated in YPD medium, the viability of those cells cultured in Sabouraud medium was only 52%. However, this reduction in viability could be prevented by the addition of 4.15 mM $MgSO_4$, with 100% viability being restored. Furthermore, the reduction in the viability of the cells cultured in Sabouraud medium was confirmed by staining the cells with SYTOX green, which penetrates only damaged cell membranes (47) (Fig. S4A).

**$MgSO_4$ supplementation accelerates hyphal growth in *T. asahii*.** We measured the cell length after culture for 16 h when the effect of $MgSO_4$ was investigated. However, previous studies have estimated hyphal growth based on colony formation on agar plates (48, 49), which takes 5 to 7 days. Instead, we investigated the change in length of JCM 2466 cells during continuous culture for 6 days in Sabouraud medium with and without 4.15 mM $MgSO_4$. During the 6 days, we measured lengths of live cells at time points of 0, 8, 24, 48, 72, 96, 120, and 144 h under the microscope. The average $OD_{660}$ value of JCM 2466 cells increased gradually and reached 0.63 at 144 h in Sabouraud medium (Fig. 3A). Thus, the rate of increase in $OD_{660}$ values was $4.38 \times 10^{-3}$/h, whereas the average $OD_{660}$ value of JCM 2466 cells increased to 0.798 at 24 h of culture in Sabouraud+Mg medium (Fig. 3A). Thus, the rate of increase in $OD_{660}$ values was $33.3 \times 10^{-3}$/h. Therefore, the growth rate of JCM 2466 cells was accelerated by 7.6 times upon adding 4.15 mM $MgSO_4$. In addition, the average lengths of JCM 2466 cells increased to 68.16 $\mu$m at 144 h when cultured in Sabouraud medium (Fig. 3B; supplemental table). The rate of increase in cell elongation was 0.441 $\mu$m/h. The cells cultivated in Sabouraud medium were small and contained lipid droplets until 48 h. However, the number of elongated and branched hyphal cells increased in Sabouraud medium between 72 and 144 h while retaining lipid droplets (Fig. 3D). In contrast, when 4.15 mM $MgSO_4$ was added to Sabouraud medium, the average length of JCM 2466 cells increased to 88.22 $\mu$m at 48 h and then decreased to 60.16 $\mu$m at 144 h (Fig. 3B; supplemental table). Cells cultivated in a medium containing $MgSO_4$ had already started hyphal growth at 8 h, which continued until 48 h (Fig. 3B and D). After 48 h, small and round cells, including arthroconidia, were produced that

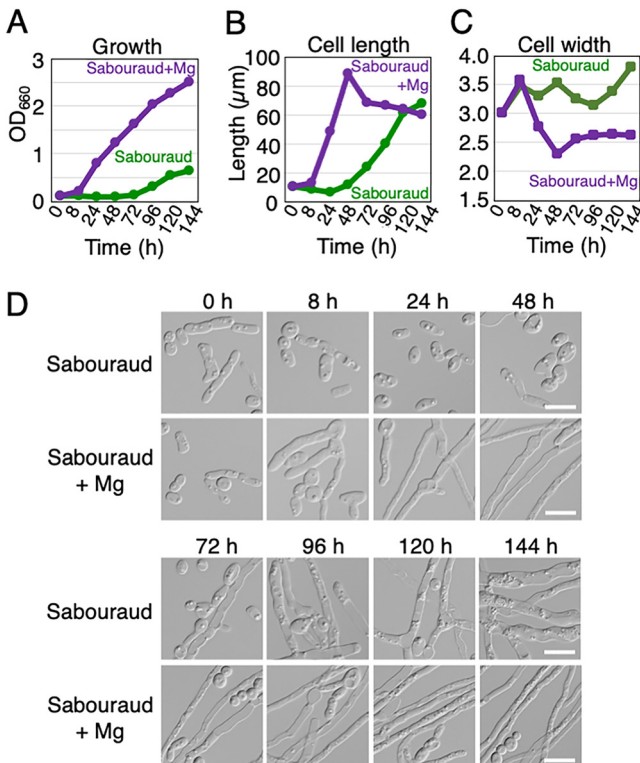

**FIG 3** Magnesium sulfate treatment accelerated hyphal growth in *T. asahii*. (A) The growth rate, (B) length, and (C) width of JCM 2466 cells cultivated in Sabouraud medium (green) and Sabouraud medium containing 4.15 mM MgSO$_4$ (purple) were measured at 8 h and every 24 h for 144 h at 25°C. The average length at each time point is given in panel B. The average width at each time point is shown in panel C. (D) The cell morphological phenotype of JCM 2466 cells is shown at each time point. The scale bar is 10 $\mu$m. OD$_{660}$, optical density at 660 nm.

consequently reduced the average cell length (Fig. 3D). The rate of increase in cell elongation from 8 h to 48 h was 1.879 $\mu$m/h. Comparison of the cell elongation phase under both conditions of Sabouraud medium with and without MgSO$_4$ showed that the rate of increase in cell elongation was 4.26 times higher upon addition of 4.15 mM MgSO$_4$ (Fig. 3B; supplemental table). Due to the acceleration in hyphal growth, the cell length reached 48.65 $\mu$m at 24 h of culture when MgSO$_4$ was added. Without MgSO$_4$, a comparable length (40.27 $\mu$m) was attained at 96 h of culture. The timing of elongation of JCM 2466 cells was hastened by approximately 72 h by adding 4.15 mM MgSO$_4$ to Sabouraud medium (Fig. 3B; supplemental table).

We also investigated changes in the width of JCM 2466 cells during a 6-day continuous culture in Sabouraud medium with or without 4.15 mM MgSO$_4$. It was observed that the average width of JCM 2466 cells gradually increased from 3.02 $\mu$m at 0 h to 3.81 $\mu$m at 144 h when cultured in the unsupplemented Sabouraud medium (Fig. 3C; supplemental table), with cell width increasing at a rate of 0.549 $\times$ 10$^{-2}$ $\mu$m/h. In contrast, following the addition of 4.15 mM MgSO$_4$ to the Sabouraud medium, the average width of JCM 2466 cells increased from 3.02 $\mu$m at 0 h to 3.57 $\mu$m at 8 h, but subsequently declined to 2.31 $\mu$m when measured at 48 h (Fig. 3C; supplemental table), with cell width decreasing at a rate of 3.15 $\times$ 10$^{-2}$ $\mu$m/h. The timing of the reduction in cell width corresponded to that of the increase in cell length at 48 h when cells were cultured in the Sabouraud+Mg medium (Fig. 3B and C). At 144 h, the average width of JCM 2466 cells had increased to 2.63 $\mu$m (Fig. 3C; supplemental table). Therefore, the width of the JCM 2466 cells cultivated in Sabouraud medium was 1.45-fold wider than that of cells cultivated in Sabouraud+Mg medium (Fig. 3B; supplemental table).

**Hyphal growth is maintained in cultures of Sabouraud medium.** We examined whether JCM 2466 cells could still respond to the addition of MgSO$_4$ when cells in the

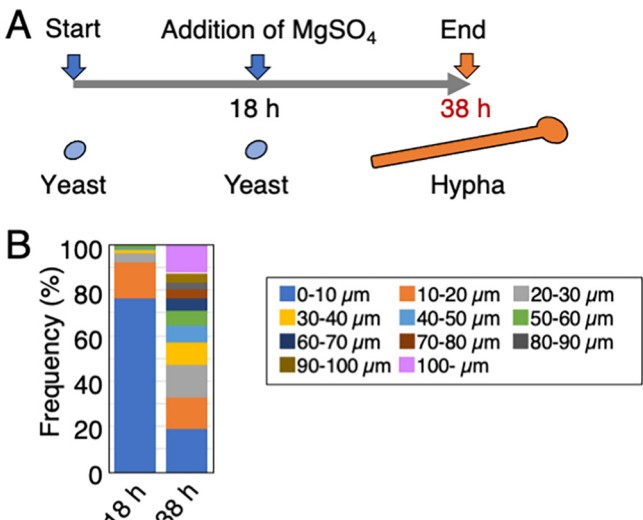

**FIG 4** *T. asahii* cells retained the ability to form hyphae even after cultivation in Sabouraud medium. To determine whether JCM 2466 cells retain the ability to respond to MgSO₄ addition after continuous cell culture in Sabouraud medium, cells were cultivated for 18 h at 25°C in Sabouraud medium, followed by further incubation for 20 h in Sabouraud+Mg medium (A) Scheme of culture of JCM 2466 cells in Sabouraud medium during 38 h. (B) Cell lengths after incubation for 18 and 38 h were measured and shown as the cell length classification. MgSO₄, magnesium sulfate.

yeast form were cultivated in Sabouraud medium. JCM 2466 cells were cultivated for 18 h in Sabouraud medium, followed by further incubation for 20 h in Sabouraud+Mg medium (Fig. 4A). We observed the cultured cells at 18 and 38 h and compared their lengths. The average cell length at 38 h (49.07 $\mu$m) was significantly longer than that at 18 h (9.49 $\mu$m) ($P < 10^{-5}$) (supplemental table). At 38 h, we observed a one-third reduction in the number of cells with a length of <20 $\mu$m and detected the appearance of cells with a length of ≧100 $\mu$m in the cell length classification (Fig. 4B; supplemental table). Therefore, we established that JCM 2466 cells could still respond to the addition of MgSO₄ and produce hyphae even after 18 h of culture in Sabouraud medium.

**Large lipid droplets are resolved in hyphae of *T. asahii*.** To determine the change in cell morphology upon addition of MgSO₄, we first investigated the size of lipid droplets, which are the markers of nutrient-starved cells (Fig. 1A). We cultivated JCM 2466 cells for 16 h at 25°C in Sabouraud medium with and without 4.15 mM MgSO₄ and stained the cells with Nile red (Fig. 5A). The large lipid droplets that appeared in the cells cultivated in Sabouraud medium diminished in size upon addition of 4.15 mM MgSO₄, but the DAPI signal did not (Fig. 5A). Moreover, fluorescence generated upon Nile red staining was dispersed within the cell when 4.15 mM MgSO₄ was added (Fig. 5A), which was similar to the case observed in YPD medium (Fig. 1A). Therefore, MgSO₄ addition influenced the morphology of lipid droplets in *T. asahii*.

**Vacuoles are enlarged in *T. asahii* depending on magnesium supplementation.** To further examine the morphological characteristics of hyphal growth in *T. asahii*, we observed the vacuoles of JCM 2466 cells because hyphae accumulate vacuoles (5, 50). We cultured JCM 2466 cells for 16 h in Sabouraud and Sabouraud+Mg media and stained the vacuoles using FM4-64. We found that small vacuoles and condensed particles were frequently observed in JCM 2466 cells cultivated in Sabouraud medium (Fig. 5B). However, the vacuoles showed extreme growth in the hyphae of JCM 2466 cells when 4.15 mM MgSO₄ was added to Sabouraud medium (Fig. 5B). The presence of vacuoles was also confirmed in the DIC images because they were transparent (Fig. 5B). We compared the sizes of vacuoles in cells cultured in Sabouraud and Sabouraud+Mg media by examining the area proportional to the difference in volume. For the purpose of the comparison, we measured the total vacuolar area in 86 cells, with vacuoles being classified by area for comparison in similar-sized cells. Vacuoles with an area of <10 $\mu$m² were detected even in small cells with an area of <50 $\mu$m² and for cells cultured in both Sabouraud and

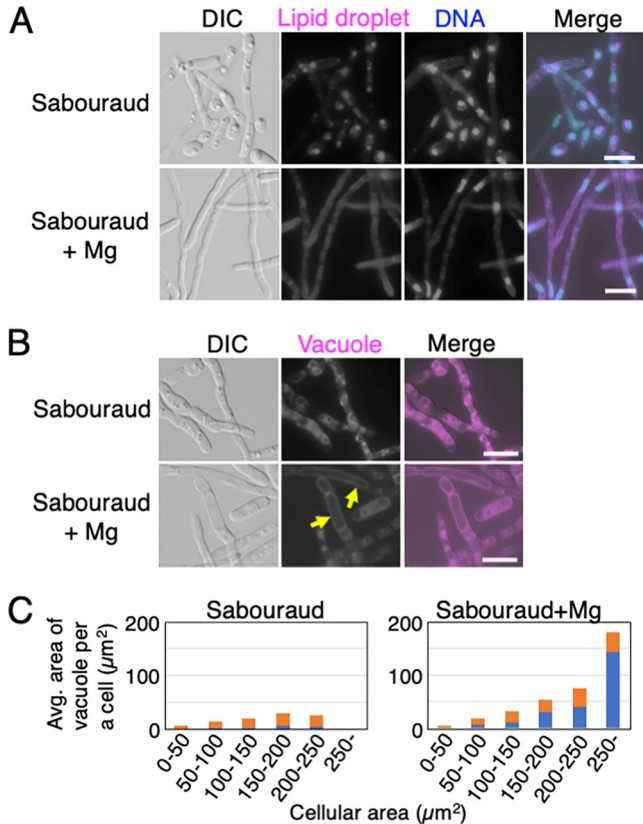

**FIG 5** Large lipid droplets were resolved in hyphae of *T. asahii*. (A) Lipid droplets and DNA of JCM 2466 cells cultivated in Sabouraud medium and Sabouraud medium containing 4.15 mM MgSO$_4$ for 16 h at 25°C were stained using Nile red and the DAPl, respectively. (B) Vacuoles in JCM 2466 cells were stained with FM4-64 after cultivation in Sabouraud medium and Sabouraud medium containing 4.15 mM MgSO$_4$ for 16 h at 25°C. Yellow arrows indicate growing vacuoles. (C) The average area of all vacuoles in a cell was measured. Vacuoles with an area of ≥10 $\mu$m$^2$ are indicated by a blue bar. Vacuoles with an area of <10 $\mu$m$^2$ are indicated by an orange bar. Vacuolar area was classified based on cellular area. The numbers of cells in the respective size (area) ranges are as follows: in Sabouraud medium, 35 cells of 0 to 50 $\mu$m$^2$, 17 cells of 50 to 100 $\mu$m$^2$, 16 cells of 100 to 150 $\mu$m$^2$, 12 cells of 150 to 200 $\mu$m$^2$, 6 cells of 200 to 250 $\mu$m$^2$, and 0 cells of greater than 250 $\mu$m$^2$; in Sabouraud+Mg medium, 19 cells of 0 to 50 $\mu$m$^2$, 13 cells of 50 to 100 $\mu$m$^2$, 14 cells of 100 to 150 $\mu$m$^2$, 14 cells of 150 to 200 $\mu$m$^2$, 10 cells of 200 to 250 $\mu$m$^2$, and 16 cells of greater than 250 $\mu$m$^2$. The scale bar is 10 $\mu$m. DIC, differential interference contrast microscopy; YPD, yeast extract-peptone-dextrose medium.

Sabouraud+Mg media, the vacuoles increased in size with an increase in cellular area (Fig. 5C). However, whereas for the cells cultured in Sabouraud+Mg medium, the number of vacuoles with an area of ≥10 $\mu$m$^2$ increased as cellular area increased, this was not observed in cells cultured in Sabouraud medium (Fig. 5C). In cells ranging in size from 150 to 200 $\mu$m$^2$, we observed a comparable number of vacuoles of area <10 $\mu$m$^2$ in cells cultivated in Sabouraud and Sabouraud+Mg media. In contrast, vacuoles with an area of ≥10 $\mu$m$^2$ were found to be 6.25-fold more abundant in cells cultivated in Sabouraud+Mg medium than in those cultivated in Sabouraud medium (Fig. 5C; supplemental table). To examine the correlation between vacuolar extension and cell viability, we stained the cells cultivated in Sabouraud medium using FM4-64 and SYTOX green. The vacuolar area was 3.1 times larger in SYTOX-negative cells than in SYTOX-positive cells ($P < 10^{-5}$) (Fig. S4B and supplemental table). Therefore, the addition of MgSO$_4$ had the effect of promoting the size of vacuoles in *T. asahii*.

**Hyphal growth in *T. asahii* is inhibited via treatment with an actin inhibitor.** To investigate which of the factors motor protein, microtubule, or actin mainly contributes to hyphal growth in *T. asahii*, we compared cell lengths of JCM 2466 cells that were treated with 100 $\mu$g/mL benzimidazole (a microtubule polymerization inhibitor) and 10 $\mu$g/mL latrunculin A (an actin polymerization inhibitor). We cultivated JCM 2466 cells with the

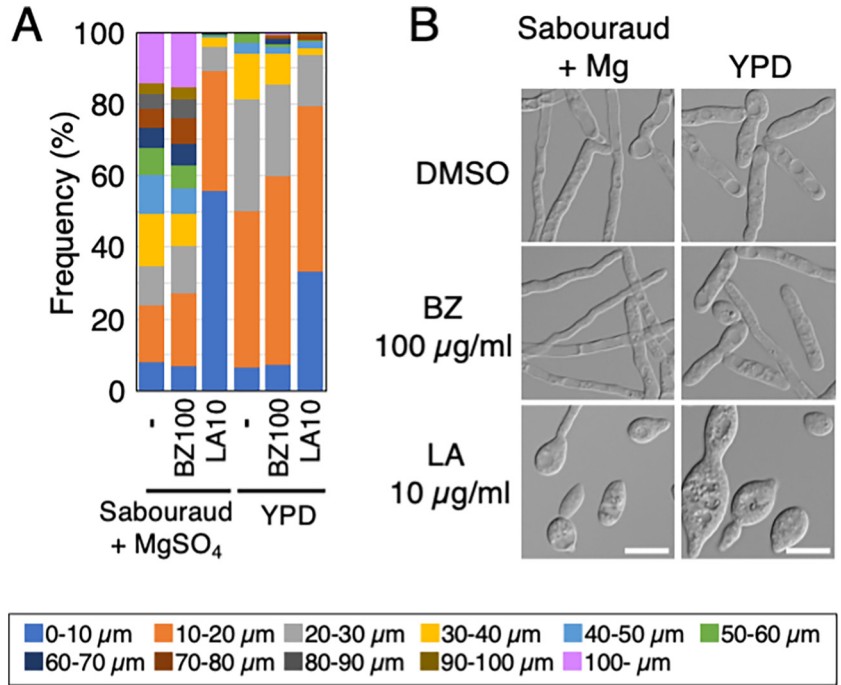

**FIG 6** Hyphal growth was inhibited in *T. asahii* by treatment with an actin inhibitor. (A) Lengths of JCM 2466 cells were measured after treatment with 100 $\mu$g/mL benzimidazole, 10 $\mu$g/mL latrunculin A, or DMSO ($-$) in YPD medium and Sabouraud medium containing 4.15 mM MgSO$_4$ and then shown as the cell length classification. (B) Morphological phenotypes observed in experiments in panel A. The scale bar is 10 $\mu$m. BZ, benzimidazole; LA, latrunculin A; YPD, yeast extract-peptone-dextrose.

inhibitors or dimethyl sulfoxide (DMSO) for 16 h in Sabouraud+Mg and YPD media and then measured the cell length. The average length of cells treated with benzimidazole was 56.17 $\mu$m, comparable to that of cells treated with DMSO (56.32 $\mu$m; $P = 0.98$); however, in cells treated with latrunculin A, the average cell length was 11.96 $\mu$m in Sabouraud+Mg medium ($P < 10^{-5}$; supplemental table). In response to the addition of latrunculin A to Sabouraud+Mg medium, the number cells with a length of <20 $\mu$m increased by approximately 4-fold, whereas we were unable to observe any cells with a length of ≥100 $\mu$m in the cell length classification (Fig. 6A; supplemental table). Similarly, the length of cells cultivated in YPD containing latrunculin A (15.87 $\mu$m) was significantly shorter than that of cells treated with DMSO (22.39 $\mu$m) or benzimidazole (22.59 $\mu$m) ($P < 10^{-5}$) (supplemental table). In response to the addition of latrunculin A to the YPD medium, there was an approximately 1.5-fold increase in the number of cells with a length of <20 $\mu$m in the cell length classification (Fig. 6A; supplemental table). The morphological phenotype was dramatically altered from hyphal form to yeast form upon treatment with 10 $\mu$g/mL latrunculin A, but not with 100 $\mu$g/mL benzimidazole (Fig. 6B). Therefore, latrunculin A treatment was more detrimental than benzimidazole to hyphal growth of *T. asahii*, suggesting that actin mainly contributes to hyphal growth in *T. asahii*.

**The distribution of mitochondria throughout the cell cytoplasm and adjacent to the cell walls in *T. asahii* is dependent on magnesium supplementation.** We examined whether the mitochondrial distribution in *T. asahii* was altered upon adding MgSO$_4$, because mitochondria interact with actin in several organisms (29, 30). We cultured JCM 2466 cells for 16 h in YPD, Sabouraud, and Sabouraud+Mg media and then stained the mitochondria using MitoBright reagent, the cell walls using calcofluor white reagent, and lipid droplets using 4,4-difluoro-4-bora-3a,4a-diaza-s-indacene (BODIPY) reagent. Mitochondria were distributed throughout the cell cytoplasm in hyphal and yeast-like cells in Sabouraud+Mg medium (Fig. 7A and B). Among the cells cultured in Sabouraud+Mg medium, a proportion of the mitochondria was observed to be distributed adjacent to the cell walls, detected using calcofluor white (Fig. 7A and B).

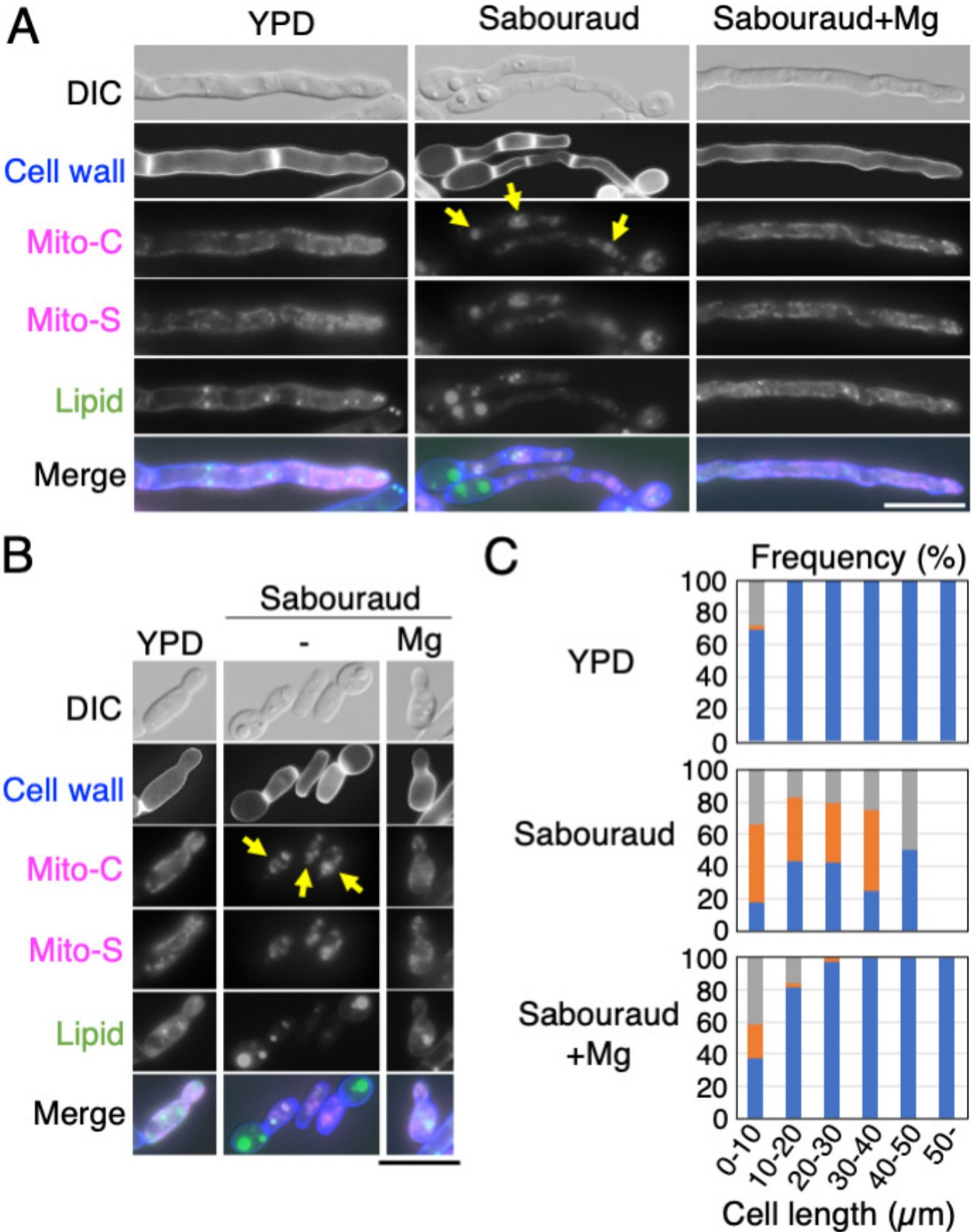

**FIG 7** Mitochondria were distributed throughout the cell cytoplasm and adjacent to the cell walls in *T. asahii* and depended on magnesium supplementation. Mitochondria, lipid droplets, and the cell walls were stained with MitoBright, BODIPY, and calcofluor white, respectively, in JCM 2466 cells cultivated for 16 h at 25°C in YPD medium, Sabouraud medium, and Sabouraud medium containing 4.15 mM MgSO$_4$. Hyphal cells are described in panel A, and yeast cells are described in panel B. Mito-C and Mito-S images indicate mitochondria distributions in the center and surface of a cell, respectively. The Mito-C images were used to produce the merged images. Arrows indicate the fragmented mitochondria in

We detected very few SYTOX-positive cells in Sabouraud+Mg medium (Fig. S4C). However, when cells were cultivated in Sabouraud medium, we observed a marked disruption of the mitochondrial distribution (Fig. 7A and B); in these cells, mitochondria were fragmented and located in the cytoplasm, or the mitochondrial distribution was decreased (Fig. 7A and B). The fragmented mitochondria did not colocalize with the lipid droplets detected using BODIPY (Fig. 7A and B). We classified the distribution of mitochondria into the following three phenotypes: (i) throughout the cell cytoplasm and adjacent to the cell walls, (ii) fragmented within the cytoplasm, and (iii) a reduced distribution (Fig. 7C). We found that approximately 45.4% of the cells grown in Sabouraud medium were characterized by mitochondria with a fragmented morphology, which was observed in both hyphal and yeast cells. In contrast, among the cells cultured in Sabouraud+Mg medium, this phenotype was mainly detected in the cells with a length of <10 $\mu$m (Fig. 7C). Among the cells grown in Sabouraud medium showing fragmented mitochondria, 74.6% were short cells with a length of <10 $\mu$m (Fig. 7C; supplemental table), and among these cells, 36.4% were characterized by an unequal segregation of mitochondria and lipid droplets (Fig. 7B), which was not observed in the cells cultured in Sabouraud+Mg medium. When an unequal segregation of mitochondria and lipid droplets occurred, 60.5% of the cells were stained by SYTOX green ($n$ = 38) (Fig. S4D). Moreover, the fragmented mitochondria were distributed even in the ghost cells lacking cytoplasm (Fig. S5). The ghost cells were produced by an unequal segregation (Fig. S5) and accounted for 35.2% of the cells with a length of <10 $\mu$m showing the fragmented mitochondria. In YPD medium, mitochondria were distributed throughout the cell cytoplasm and adjacent to the cell walls (Fig. 7A and B); however, we detected very few fragmented mitochondria and SYTOX-positive cells in the cells cultivated in this medium (Fig. 7C; Fig. S4E).

When the observation of a z-section under the microscope was shifted from the cell center to the surface, the localization of mitochondria was detected as dotted lines in the cells cultivated in Sabouraud+Mg and YPD media (Fig. 7A and B). To further confirm the distribution of mitochondria in three dimensions, confocal laser scanning microscopy was performed to observe mitochondria in regions where vacuoles were not stained by FM4-64. Cell images were stacked along the *z* axis, and then a cross section of it along a vertical orange line and horizontal orange line was observed from the side. Mitochondrial signals in the side view formed a ring shape in the cells cultivated in YPD (Fig. 8A) and Sabouraud+Mg media (Fig. 8C), but not in Sabouraud medium (Fig. 8B). These results indicated that mitochondria were distributed adjacent to the cell walls in *T. asahii* when magnesium was added.

**Mitochondrial distribution is altered by treatment with an actin inhibitor.** To determine whether the mitochondrial distribution in *T. asahii* was affected by an actin inhibitor, we observed mitochondria 16 h after adding latrunculin A or DMSO to cell cultures in YPD, Sabouraud, and Sabouraud+Mg media. The mitochondrial distribution was disrupted by adding latrunculin A and changed to dot-like distribution in the swollen cells in YPD and Sabouraud+Mg media (Fig. S6). To further determine whether an actin inhibitor disrupted the distribution of mitochondria in hyphal cells, we examined mitochondria 5 h after treating the cells with either latrunculin A or DMSO and compared them to cells cultivated for 16 h in YPD, Sabouraud, and Sabouraud+Mg media. Among the cells cultivated in YPD and Sabouraud+Mg media, the addition of latrunculin A was observed to disrupt the distribution of mitochondria, producing a dot-like pattern within the cytoplasm, even in hyphal cells (Fig. 9A and C) whereas mitochondrial distribu-

**FIG 7** Legend (Continued)
cells grown in Sabouraud medium. (C) Mitochondrial distribution was classified into following three phenotypes: distribution throughout the cell cytoplasm and adjacent to the cell walls (blue bar), fragmented distribution (orange bar), and reduced distribution (gray). The cell numbers in each size (length) range were as follows: in YPD medium: 32 cells of 0 to 10 $\mu$m, 134 cells of 10 to 20 $\mu$m, 92 cells of 20 to 30 $\mu$m, 20 cells of 30 to 40 $\mu$m, 5 cells of 40 to 50 $\mu$m, and 2 cells of greater than 50 $\mu$m; in Sabouraud medium, 182 cells of 0 to 10 $\mu$m, 53 cells of 10 to 20 $\mu$m, 19 cells of 20 to 30 $\mu$m, 4 cells of 30 to 40 $\mu$m, 2 cells of 40 to 50 $\mu$m, and 0 cells of greater than 50 $\mu$m; in Sabouraud+Mg medium, 80 cells of 0 to 10 $\mu$m, 49 cells of 10 to 20 $\mu$m, 34 cells of 20 to 30 $\mu$m, 24 cells of 30 to 40 $\mu$m, 21 cells of 40 to 50 $\mu$m, and 28 cells of greater than 50 $\mu$m.

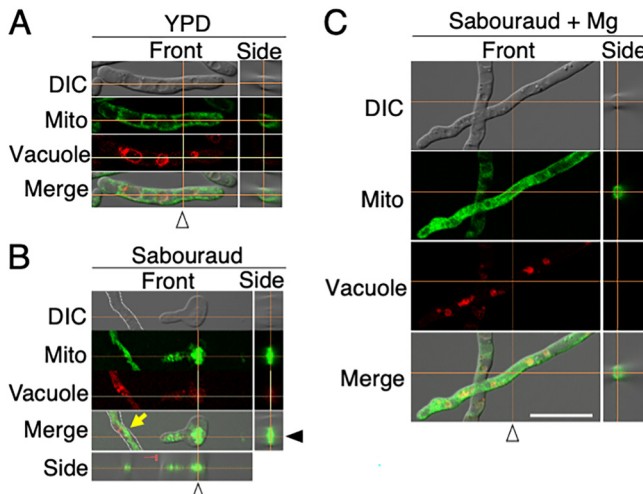

**FIG 8** Mitochondrial distribution observed by confocal scanning microscopy. Mitochondria and vacuoles of JCM 2466 cells cultivated in (A) YPD medium, (B) Sabouraud medium, and (C) Sabouraud medium containing 4.15 mM MgSO$_4$ for 16 h at 25°C were stained with MitoBright and FM4-64, respectively, and then observed under a confocal scanning microscope. White arrowheads indicate a vertical orange line showing the position of the cell cross section for the side view. A black arrowhead indicates a horizontal orange line showing the position of the hyphal cell cross-section for the side view. A hyphal form in Sabouraud medium is indicated by an arrow. A side view is shown of the hyphal form attached under the merged image. The dotted lines indicate a medial plane of a hyphal cell. The scale bar is 10 $\mu$m. YPD, yeast extract-peptone-dextrose; DIC, differential interference contrast microscopy.

tion in cells cultivated in Sabouraud medium was not affected by latrunculin A treatment (Fig. 9B). When latrunculin A was added, the dot-like pattern of mitochondrial distribution was observed in 99.1% of cells cultivated in YPD medium and in 96.3% of cells cultivated in Sabouraud+Mg medium, whereas the pattern was not observed in the media treated with DMSO. In addition, the swollen cells were abundant when the cells were cultivated for 16 h with latrunculin A (Fig. S6). The dot-like distribution of the mitochondria did not colocalize with the distribution of lipid droplets detected using BODIPY staining, and no signals were detected in unstained cells (Fig. 9). Therefore, the mitochondrial distribution was associated with actin polymerization.

## DISCUSSION

Here, we developed a magnesium-dependent hyphal formation (MagHF) method to study the yeast-to-hyphal form transition mechanism in *T. asahii*, a dimorphic basidiomycete. To develop this method, we examined hyphal formation in response to different compounds present in YNB and found that Mg$^{2+}$ was a key cation required for hyphal growth in *T. asahii*. Namely, Mg$^{2+}$ uptake induced hyphal formation most often compared to that induced by other cations, including K$^+$, Na$^+$, Ca$^{2+}$, Mn$^{2+}$, Zn$^{2+}$, Fe$^{3+}$, and Cu$^{2+}$ (Fig. 2A and B). This result is similar to a previous finding on *C. albicans* in which hyphal formation occurs more frequently upon addition of Mg$^{2+}$ than upon addition of Mn$^{2+}$, Ca$^{2+}$, and Zn$^{2+}$ (23). A shortage of magnesium inhibits morphological transition, biofilm formation, and epithelial cell adherence in *C. albicans* (51). The method in this study allowed the induction of hyphae only by adding magnesium compound to Sabouraud liquid medium comprising 4% glucose and 1% Bacto peptone. Bacto peptone is suitable for use in this method as a limiting nutrient that highlights the significance of magnesium. However, the ability to produce hypha is maintained in cells continuously cultivated for 18 h in Sabouraud medium, which indicates that the cells can respond to the addition of Mg$^{2+}$ for 18 h in Sabouraud medium (Fig. 4B). Bacto peptone contains a minimum number of nutrients, including Mg$^{2+}$, which can contribute to the maintenance of hyphal formation in *T. asahii*. An increase in Mg$^{2+}$ concentration upon addition of 4.15 mM MgSO$_4$ accelerates the process of hyphal extension. In addition, the length of cells cultivated in YPD medium and Sabouraud medium with YNB was approximately 3 times

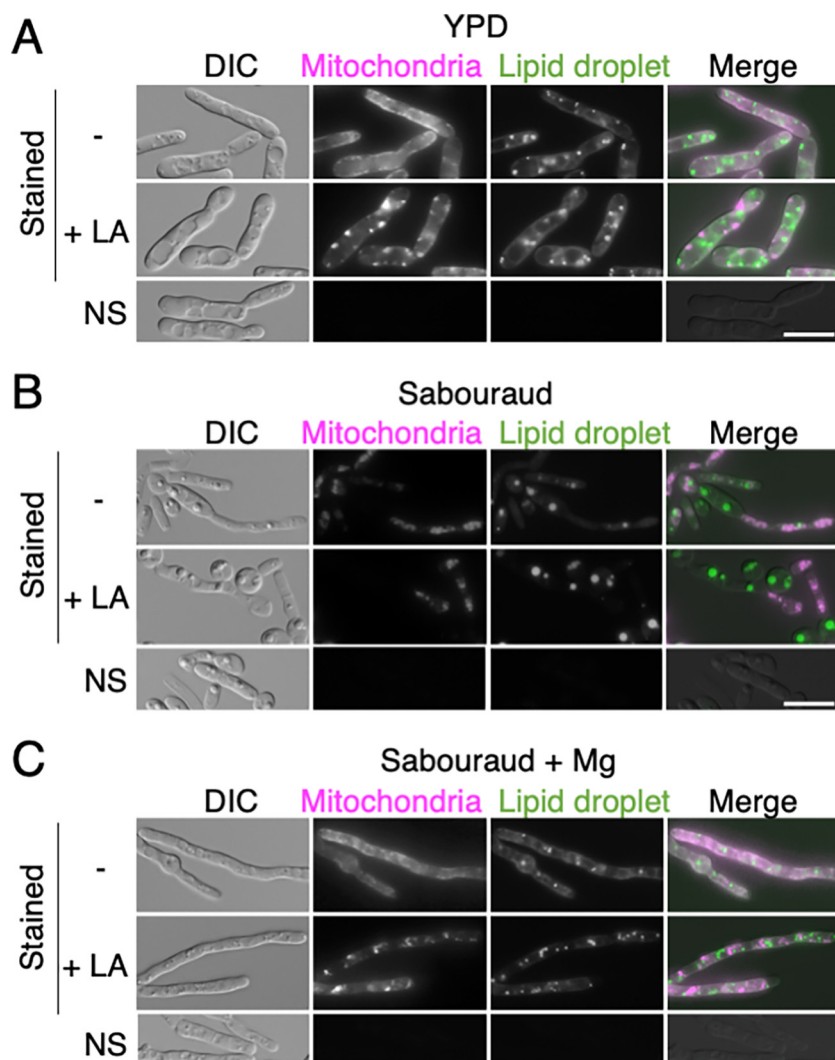

**FIG 9** Mitochondrial distribution was disrupted via treatment with actin inhibitor. JCM 2466 cells cultivated in (A) YPD medium, (B) Sabouraud medium, and (C) Sabouraud medium containing 4.15 mM MgSO$_4$ for 16 h at 25°C were treated with 10 $\mu$g/mL latrunculin A or DMSO ($-$) and cultivated for 5 h at 25°C. Subsequently, mitochondria and lipid droplets were stained using MitoBright and BODIPY, respectively. NS, not-stained control. The scale bar is 10 $\mu$m. LA, latrunculin A; DIC, differential interference contrast microscopy; DMSO, dimethyl sulfoxide.

shorter than that of cells cultivated in Sabouraud+Mg medium, suggesting that YNB contains some nutrients that shorten the *T. asahii* cells. In the future studies, simultaneous addition of the nutrient with MgSO$_4$ should be investigated to synergistically improve the outcomes of the MagHF method.

Using this method, initial hyphae, and not mature hyphae, can be observed (Fig. 3B). The low abundance of arthroconidia may be attributed to the cultivation for only 16 h, which was completely different from the traditional colony formation assay, in which colonies are cultured for 5 to 7 days on an agar plate, where arthroconidia are produced frequently. As a consequence of the low abundance of arthroconidia, we observed the production of initial hyphae until 48 h (Fig. 3B and C). In mature hyphae after 48 h, the cells cultivated in Sabouraud medium were 1.45-fold wider than those cultured in Sabouraud+Mg medium, suggesting that an increase of Mg$^{2+}$ may restrict cell width. It is advantageous that *T. asahii* transitions from yeast to hyphal form during vegetative and not sexual generation; thus, an inherent nature of hyphal growth is expected to be uncovered. This is different from the case of *C. neoformans*, a well-known basidiomycete,

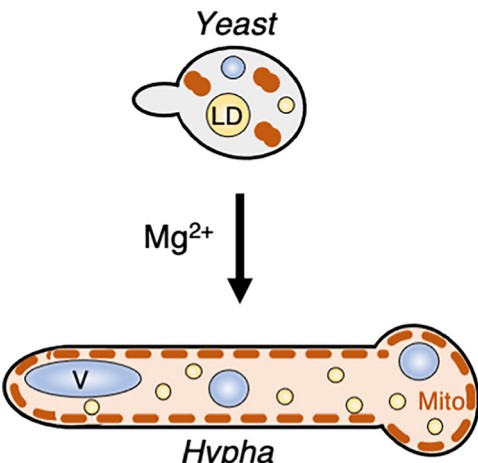

**FIG 10** Model of the effects of magnesium on hyphal cell development. Shown are illustrations of *T. asahii* cells cultivated in Sabouraud medium with or without 4.15 mM MgSO$_4$. In the cells cultivated in Sabouraud medium, there is an accumulation of large lipid droplets and fragmented mitochondria. In response to an increase in the concentration of Mg$^{2+}$, *T. asahii* cells elongate and form hyphae. Within the hyphal cells, large lipid droplets are resolved, vacuoles are enlarged, and mitochondria are distributed throughout the cell cytoplasm and adjacent to the cell walls. V, vacuoles; LD, lipid droplets; Mito, mitochondria.

in which hyphal growth is closely related to the mating process during sexual genera- tion (52). However, in our method, it is difficult to distinguish between yeast cells and small cells separated from the arthroconidia, because both cell types are small (the length is ~10 $\mu$m) and separated cells from arthroconidia may bud out as a yeast cell does. Future studies should attempt to discriminate between the two cell types. By ana- lyzing hyphae via this method, the mechanisms via which yeast cells transform into hyphae and how organelles adapt to the hyphal form in basidiomycetes can be eluci- dated. Meanwhile, the length of cells grown in Sabouraud medium increased slightly fol- lowing CaCl$_2$ supplementation (Fig. 2B). An increase in cell length upon calcium supple- mentation was also reported in *C. neoformans* (53), *C. albicans* (54), and *N. crassa* (55). The calcium supply regulates actin polymerization (56, 57) and microtubule polymeriza- tion (58). However, in the method used for *T. asahii*, calcium supplementation had less effect than magnesium supplementation.

Hyphae extended in a polar direction upon addition of MgSO$_4$ in *T. asahii*. The hyphal growth was inhibited upon addition of latrunculin A (Fig. 6B). The inhibited cells were shortened and showed a swollen phenotype that may be caused by the deficiency of a polar direction and the incubation time with latrunculin A. The phenotype is similar to the case of latrunculin A-treated cells showing a round shape without germ tubes in *C. albicans* (59) and the case of a deletion mutation in *rodZ* that is required for the determi- nation of rod shape in *Escherichia coli* (60). Therefore, actin polymerization may lie down- stream in a starvation-associated signaling pathway (4) affected by Mg$^{2+}$ in *T. asahii*. However, our findings may indicate another possibility—that Mg$^{2+}$ uptake increases the volume, although not the length, of *T. asahii* cells—as we observed that under condi- tions of actin inhibition, the volume of cells grown in Sabouraud+Mg medium was larger than of cells cultured in Sabouraud medium (Fig. S6). In this case, the starvation- associated signaling pathway may be independent of actin polymerization. In both the cases, actin polymerization may contribute mainly to the growth of hyphae in *T. asahii*.

The mechanism by which magnesium contributes to hyphal growth is not fully eluci- dated in the study. The length of *T. asahii* cells gradually shortened upon addition of MgSO$_4$ solutions at different concentrations (Fig. 2D). Thus, morphological phenotypes might not be caused by changes associated with Mg$^{2+}$ and magnesium transporters that allosterically regulate the influx of Mg$^{2+}$ through the plasma membrane (61–63). We observed several morphological deficiencies, including defective growth, reduced cell

length, larger lipid droplets, and fragmented distribution of mitochondria, in cells cultured in Sabouraud liquid medium (Fig. 10). These phenotypes were suppressed following magnesium supplementation (Fig. 2B, Fig. 5A and B, and Fig. 7A). Particularly, the fragmented distribution of mitochondria was restored to a normal distribution throughout the cell cytoplasm and adjacent to the cell walls (Fig. 7A and B). This phenotypic change indicated that magnesium intake is associated with the regulation of mitochondrial distribution in *T. asahii*. The fragmented distribution of mitochondria in hyphal cells was also observed upon the addition of latrunculin A (Fig. 9A and C), suggesting that the mitochondrial distribution in *T. asahii* is associated with the actin cytoskeletal network in *T. asahii*, as shown in higher eukaryotic cells (30). The fragmented distribution of mitochondria may be independent of the difference in cell length and may be a patterned distribution rather than a defective distribution in hyphal cells. A loss of cell viability in Sabouraud medium may be attributed to the fragmentation of mitochondria in short cells. The fragmented mitochondrial phenotype was associated with an unequal segregation of mitochondria and lipid droplets (Fig. 7B; Fig. S4D), and thus the segregation of mitochondria and lipid droplets may be inhibited in Sabouraud medium. In *S. cerevisiae*, it has been found that a fragmented distribution of mitochondria is associated with a reduction in respiration (43). During migration in higher eukaryotic cells, mitochondria interact with the actin cytoskeleton throughout the cell and release products generated during respiration (for example, NADH) to the actin cytoskeleton (64). Although in the present study we failed to establish a direct association between actin and mitochondria, the mitochondrial distribution was observed to be disrupted in the presence of an actin inhibitor (Fig. 9; Fig. S6). Thus, establishing the mechanisms whereby actin maintains a normal mitochondrial distribution via $Mg^{2+}$-related signaling will contribute to gaining a better understanding of hyphal formation in *T. asahii*.

Although lipids are rarely detected in JCM 2466 cells, we observed that large lipid droplets are produced in the cytoplasm of these cells when cultivated in Sabouraud medium (Fig. 1A, Fig. 5A, and Fig. 7A). The phenotype of large lipid droplets in JCM 2466 cells was similar to that of cells in which growth was stalled in a nitrogen-deficient medium in the red alga *Cyanidioschyzon merolae* (65). Lipid droplets may not disperse throughout the cells cultivated in Sabouraud medium; therefore, nutrients may not be supplied to the mitochondria. The mechanisms whereby lipid droplets accumulate in cells cultured in Sabouraud medium might be the key to understanding hyphal growth in *T. asahii*.

Moreover, we found that the number of large vacuoles greater than 10 $\mu$m$^2$ in size, although not small vacuoles with a size of less than 10 $\mu$m$^2$, increased when $MgSO_4$ was added to Sabouraud medium, thereby indicating that an increase in $Mg^{2+}$ might induce the fusion of small vacuoles. These observations may indicate the opposite effect of magnesium deprivation in *C. albicans*, in which vacuolar homeostasis was found to be abrogated by disrupting morphology and acidification (66). In *C. albicans*, FM4-64 has been shown to be localized to the vacuolar lumen and diffusely distributed under conditions of magnesium deprivation (66). Consequently, uptake of $Mg^{2+}$ may enhance vacuolar homeostasis and promote hyphal growth.

Taken together, our results reveal a new method to study the yeast-to-hypha transition in *T. asahii*. Additionally, we suggest that $Mg^{2+}$ uptake influences mitochondrial distribution, the production of lipid droplets, and vacuolar growth, which contribute to promotion of hyphal growth in *T. asahii* (Fig. 10). Studying the relationships between $Mg^{2+}$ and the properties of cell organelles will be important for gaining a better understanding of the transition of yeast cells to a hyphal form and thus how pathogenicity arises.

## MATERIALS AND METHODS

**Strains and media.** *T. asahii* JCM 2466 cells, provided by the Japan Collection of Microorganisms, RIKEN BioResource Research Center (Tsukuba, Japan), were cultivated in YPD medium (2% glucose, 1% yeast extract, and 2% Bacto peptone), Sabouraud medium (4% glucose and 1% Bacto peptone), and Sabouraud medium supplemented with each nutrient. The concentration of the nutrients was equalized to that of YNB (Difco, Sparks, MD, USA). To identify the nutrients required for hyphal growth in *T. asahii*, the content of the YNB was classified into four groups of nutrients: amino acids, trace elements, salts, and

vitamins (67, 68). The salt group included the nitrogen source in the study. The amino acid group contained L-histidine monohydrochloride, LD-methionine, and LD-tryptophan. The trace element group contained boric acid, manganese sulfate, zinc sulfate, ferric chloride, sodium molybdate, potassium iodide, and copper sulfate. The salt group contained ammonium sulfate, monopotassium phosphate, $MgSO_4$, sodium chloride, and calcium chloride. The vitamin group contained inositol, niacin, pyridoxine, thiamine, calcium pantothenate, riboflavin, *p*-aminobenzoic acid, folic acid, and biotin. To examine which motor protein, microtubule, or actin, mainly contributes to hyphal growth in *T. asahii*, 100 $\mu$g/mL benzimidazole (TGI, Tokyo, Japan) or 10 $\mu$g/mL latrunculin A (Fujifilm, Osaka, Japan) was added to the culture media.

**MagHF method.** Cells ($OD_{660}$ = 0.1) were inoculated in 5 mL Sabouraud medium with or without 4.15 mM $MgSO_4$ and incubated in a BioShaker BR-23FP air shaker (TAITEC, Koshigaya City, Japan) at 100 rpm for 16 h at 25°C. Sabouraud medium was composed of 4% glucose and 1% Bacto peptone. The $OD_{660}$ value was measured using miniphoto 518R (TAITEC). After incubation, each sample (1 mL) was transferred to a 1.5-mL tube, centrifuged at 2,260 $\times$ *g*, and the supernatant was removed leaving a medium of 20 $\mu$L. The cell sample was mixed by tapping and used for observation under a microscope Olympus BX53 under $\times$40 or $\times$100 magnification equipped with a U-FDICT mirror unit (Olympus, Tokyo, Japan). The cell length was measured between the cell tips along the cell shape using the software CellSens (Olympus). The *P* value for each cell length was estimated using the Mann-Whitney U test. A *P* value of <0.05 was considered to indicate significant difference.

**Measurement of growth rate.** Cells were inoculated in each liquid medium (5 mL) when the $OD_{660}$ was 0.05 and incubated in a bio-photorecorder TVS062CA air shaker (Advantec) at 100 rpm for 72 h or 144 h at 25°C. The $OD_{660}$ values were recorded every 10 min.

**Quantification of vacuolar area.** The area of vacuoles that were stained by FM4-64 was measured using CellSens software (Olympus). Vacuoles in cells were classified into two different size groups, namely, those of $\geq$10 $\mu$m$^2$ and those of <10 $\mu$m$^2$. The area of each group was accumulated in a cell and ranged for each cellular area. The ranges of cellular area were classified as follows: 0 to 50 $\mu$m$^2$, 50 to 100 $\mu$m$^2$, 100 to 150 $\mu$m$^2$, 150 to 200 $\mu$m$^2$, 200 to 250 $\mu$m$^2$, and greater than 250 $\mu$m$^2$.

**Colony formation assay.** Samples of cells cultivated in YPD, Sabouraud, and Sabouraud+Mg media were prepared using the MagHF method. Samples (10 $\mu$L) were transferred to a cell counting chamber (BRAND GmbH, Wertheim, Germany). Cell number was counted in a volume of 0.1 mm$^3$ under a microscope. For cells cultured in YPD medium, 172 cells were cultivated on 10 YPD agar plates for 2 days. An average of 201.4 cells were produced, indicating a viability of 117%. For cells cultured in Sabouraud medium, 253 cells were cultivated on 10 YPD agar plates for 3 days. An average of 131.3 cells were produced, indicating a viability of 52%. For cells cultured in Sabouraud+Mg medium, 181 cells were cultivated on 10 YPD agar plates for 2 days. An average of 196.1 cells were produced, indicating a viability of 108%.

**Cell staining.** The incubated cells were harvested and fixed using 70% ethanol (1 mL) for 1 h at 4°C. The cells were washed once with 1 mL phosphate-buffered saline (PBS), mixed with 1 mL PBS containing 100 nM Nile red (Fujifilm, Tokyo, Japan), and left to stand in the dark for 1 min at room temperature (25 to 28°C). Subsequently, the cells were washed twice with 1 mL PBS and suspended in 50 $\mu$L PBS. The cell suspension (1 $\mu$L) was mixed well with 2 $\mu$M DAPI (1 $\mu$L) (Dojindo, Kumamoto, Japan) on a glass slide before observation. Slides were observed using Olympus BX53 and a 100$\times$ lens objective. U-FUNA and U-FGW mirror units were used in DAPI and Nile red staining experiments, respectively (Olympus). Nile red was diluted to a 100 $\mu$M concentration in methanol, and this stock solution was stored at $-$20°C. DAPI was diluted to 50 $\mu$M in PBS, and this stock was stored at 4°C. For live cell staining, the cells were washed once with 1 mL PBS and mixed with 1 mL PBS containing 100 nM BODIPY (Thermo Fisher Scientific, Waltham, MA, USA) and 100 nM MitoBright (Dojindo) and then left to stand in the dark for 10 min at 25 to 28°C. The cells were washed once with PBS (1 mL) and then suspended in 50 $\mu$L of PBS before observation. When the cell wall was stained with calcofluor white (Sigma-Aldrich, St. Louis, MO, USA), the calcofluor white solution was 1,000-fold diluted in PBS and added to 1 mL of PBS solution containing BODIPY and MitoBright. BODIPY was diluted to a concentration of 100 $\mu$M in DMSO and stored at $-$20°C. The U-FGFP mirror unit was used (Olympus). Alternatively, for FM4-64 staining, the incubated cells (1 mL) were transferred to a 1.5-mL tube and the supernatant was removed. The cells were suspended in 49 $\mu$L of Sabouraud medium with and without 4.15 mM $MgSO_4$. The suspended cells were mixed with 2 mM FM4-64 (1 $\mu$L) (Tocris, Bristol, United Kingdom) and left to stand in the dark for 20 min at 25 to 28°C. Subsequently, the cells were washed twice with each medium (0.5 mL) and suspended in each medium (20 $\mu$L) before observation. The U-FGW mirror unit was used (Olympus). FM4-64 was diluted to 10 mM in DMSO, and this stock was stored at $-$20°C. Light was supplied using U-HGLGPS (Olympus).

**Confocal laser scanning microscopy.** Cells were inoculated in 5 mL of YPD, Sabouraud, and Sabouraud+Mg media when the $OD_{660}$ was 0.1 and incubated at 100 rpm for 16 h at 25°C. Then, the cells were stained using MitoBright green (Dojindo) and FM4-64 (Tocris, Bristol, United Kingdom), simultaneously. Confocal z-stacks (z-plane distance of 0.125 $\mu$m) were acquired with a confocal laser scanning microscope (A1R + Ni-E with NIS Elements AR software; Nikon, Tokyo, Japan) equipped with a LU-N4/N4S 4-laser unit and an Apo $\times$100/1.45 oil lens objective. The confocal z-stacks of 71 slices (8.75 $\mu$m) in a YPD sample, 51 slices (6.25 $\mu$m) in a Sabouraud sample, and 74 slices (9.13 $\mu$m) in a Sabouraud sample with 4.15 $MgSO_4$ were acquired. Samples were mounted in PBS under glass coverslips and excited with a 488-nm laser for MitoBright fluorescence (green) and 561-nm laser for FM4-64 fluorescence (red). A DM405/488/561/640 diachronic mirror and BA570-620 emission filter were used for observation. The pinhole setting was 25.5 $\mu$m for the merged DIC and fluorescence images.

## SUPPLEMENTAL MATERIAL

Supplemental material is available online only.

**SUPPLEMENTAL FILE 1**, PDF file, 11.1 MB.

## ACKNOWLEDGMENTS

We thank Ri-ichiroh Manabe, Laboratory for Comprehensive Genomic Analysis, RIKEN Center for Integrative Medical Sciences, for discussions and valuable comments.

This work was supported by the Institute for Fermentation, Osaka (IFO), Japan. We thank Editage (www.editage.com) for English language editing.

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
