## [Reviewer comments · Microbiology Spectrum]

Microbiology Spectrum

Hyphal growth in *Trichosporon asahii* is accelerated by the addition of magnesium

Keita Aoki, Kosuke Yamamoto, Moriya Ohkuma, Takashi Sugita, Naoto Tanaka, and Masako Takashima

Corresponding Author(s): Keita Aoki, Tokyo Nogyo Daigaku

Review Timeline:

Submission Date:	October 20, 2022
Editorial Decision:	November 22, 2022
Revision Received:	January 23, 2023
Editorial Decision:	February 21, 2023
Revision Received:	March 23, 2023
Accepted:	April 10, 2023

Editor: Carolina Coelho

Reviewer(s): The reviewers have opted to remain anonymous.

Transaction Report:

DOI: <https://doi.org/10.1128/spectrum.04242-22>

November 22, 2022

Dr. Keita Aoki
Tokyo Nogyo Daigaku
Tokyo NODAI Research Institute
1-1-1 Sakuragaoka
Setagaya
Tokyo, Tokyo 156-8502
Japan

Re: Spectrum04242-22 (Hyphal growth in *Trichosporon asahii* is accelerated by the addition of magnesium)

Dear Dr. Keita Aoki:

After considering the reviewer's comments, my decision is this paper requires significant revision before acceptance. In particular, I agree with reviewer 1 and 2 and agree that the data is insufficient to support the conclusions at this stage. The microscopy data and analysis of several figures needs significant improvements to show co-localization. Further the data showing of how Mg is influencing organelle distribution needs to take into account the same morphological forms, as reviewer #2 points out. The authors needs to adress the majority, if not all, of the reviewer's comments.

Link Not Available

Sincerely,

Carolina Coelho

Journals Department
Reviewer comments:

Reviewer #1 (Comments for the Author):

The study by Aoki et al. describes the identification of magnesium as a limiting factor for hyphal growth in Sabouraud medium. By exploiting the presence and absence of magnesium in the growth medium, they described defects in the lipid droplets distribution and distinct mitochondrial and vacuolar morphologies in *Trichosporon asahii*. The authors described enlarged vacuoles in media supplemented with MgSO₄. The authors also proposed that yeast cells have a fragmented mitochondrial pattern, while in hypha the morphology of mitochondria was associated with filaments of actin and organized beside of the plasm membrane. In *Candida albicans*, the deprivation of magnesium inhibits the hypha formation, cell adherence and results in diminished hyphal damage during macrophage infection (PMID: 30659981; PMID: 35834579). In accordance, the absence of the micronutrient magnesium seems to be essential to the formation of hyphal forms associated with the virulence in *Trichosporon asahii*. The manuscript is technically sound and describes an interesting finding for *Trichosporon asahii* that can be valuable for future virulence studies. However, there are points that the authors might consider:

-The authors could add arrows to highlight the lipid droplets that are not associated to DNA in figure 1.

-The figure 4A and the text do not the design of the experiment.

-The authors suggested that the vacuoles are bigger after the addition of MgSO₄ compared with YPD. While such observation is supported by the image in figure 5, the microscopy on figure 7 shows vacuoles of similar in sizes for cells grown in YPD and Sabouraud supplemented with MgSO₄. Did the authors try somehow to quantify this phenomenon of enlarged vacuoles?

-In the passage: "Therefore, actin polymerization may be downstream in a signalling pathway...", please clarify the association made between actin polymerization signalling and the findings related to YPD and magnesium-containing medium.

-Is the fragmented morphology of mitochondria observed in yeast a deficiency or a pattern? Cell ghosts also had the fragmented morphology, how was evaluated the viability of the cells cultivated in Sabouraud for the microscopy?

-If morphology of mitochondria observed in yeast is different from hypha and actin is essential for hypha formation, it seems correct that the morphology of mitochondria would be the same as observed in yeast with the treatment of latrunculin A. Would the mitochondria morphology change if, after hypha formation, it was added latrunculin A and incubated for 4-5 hours?

-In the passage: "The length of *T. asahii* cells gradually decreased..", are the cells diminishing or the higher concentrations of magnesium giving better support for hyphal growth?

-I would suggest that the authors revise their conclusions about the role of Mg²⁺, transporters and gene expression, as some of them sound very speculative and are not supported by the findings described in the manuscript.

Reviewer #2 (Comments for the Author):

Hyphal growth in *Trichosporon asahii* is accelerated by the addition of magnesium.
Aoki et al.

This paper seeks to examine factors contributing to hyphal formation of *Trichosporon asahii* as well as intracellular organisation in defined and rich media. The authors show that the addition of magnesium accelerates hyphal growth for a limited time. Distribution and morphology of lipid droplets, vacuoles, cortical actin patches, and mitochondria were examined in cells cultured in different growth media.

However, authors compare organelle distribution in yeast cells with polarised hyphal forms, attributing difference in intracellular organisation to media composition, while failing to compare the same morphological forms. Microscopy images are not sufficiently clear to serve as evidence behind the conclusions drawn.

Major comments:

1. Cell length measurements seem to be highly variable, with SD exceeding mean for many conditions. Since the authors aimed to examine hyphal induction, there could be more suitable ways to present the data - for example, in a form of a morphology index.

2. Unclear microscopy images mainly showing medial cell plane in fixed cells, while drawing conclusions on organelle co-localisation and their dynamics in cells grown in different media. For example, in Fig 7, plasma membrane staining is required to demonstrate mitochondrial distribution adjacent to the membrane. Fig 8 attempts to show association of mitochondria with actin patches, but images are not clear and no control to account for background fluorescence is shown.

3. Differences in organelle morphology and distribution could largely be explained by the lack of polarised growth and the cell cycle arrest in cells cultured in Sabouraud medium. For example in Fig 5, cells in the same growth stages need to be examined for vacuolar morphology.

4. Conclusions drawn in Results and Discussion sections regarding organelle co-localisation, mitochondrial activity, functional lipid droplets, and actin polymerisation have not been convincingly substantiated in this paper.

Minor comments:

5. Page 7, paragraph 2. Clarify whether OD600 or OD660 was used.

6. Fig 2B. Statistical analysis should compare salt drop-out media with media + salt to determine which ions were required for growth, and not with MgSO₄Δ.

7. Fig 2D. Actual concentrations should be included on the figure and in the supplemental table.
8. Fig 3C. Hyphal width looks significantly different in Sabouraud medium and Sabouraud with MgSO₄. This should be acknowledged in the results.
9. Page 11, paragraph 2. Fig 4 shows that magnesium can accelerate hyphal germination and elongation of the cells pre-incubated in Sabouraud medium. Claiming that these data show that cells retain the ability to form hyphae after 18h in Sabouraud medium is incorrect, when Fig 3 clearly shows hyphae are able to form in Sab alone.
10. Page 12 line 1. Dynamics of lipid droplets - "morphology" might be better suited for describing these differences.
11. Use of language: vitamin mix or vitamins - not vitamin in a singular form to describe combination of compounds.
12. Fig 7B: a cross section of the hyphal form should also be shown for cells grown in Sabouraud medium alone.
13. P17 line 18-22. These conclusions are confusing - consider revising.

Reviewer #3 (Comments for the Author):

The manuscript presents data on the mechanism of yeast-to-hypha transition of *Trichosporon asahii* and identifies the factors that induce its hyphal growth. The data are original and there are no reports in the literature about the factors that stimulate the dimorphism of *T. asahii*. The results found are interesting and will contribute significantly to the target scientific community. The manuscript is well written and organized.

Here are some considerations.

1. Format the list of references according to the style of the journal.
2. Figure 3: Insert the image of cells at time 0 in Sabouraud + MgSO₄ medium

Staff Comments:

Preparing Revision Guidelines

Please return the manuscript within 60 days; if you cannot complete the modification within this time period, please contact me. If you do not wish to modify the manuscript and prefer to submit it to another journal, please notify me of your decision immediately so that the manuscript may be formally withdrawn from consideration by Microbiology Spectrum.

The manuscript “Hyphal growth in *Trichosporon asahii* accelerated by addition of magnesium” is a well-conducted work, data are interesting, and the authors have contributed to increasing the knowledge of the *Trichosporon* genus over the last years. I have a few commentaries to improve the research, but considering that the submission wasn't page or line numbered, it will be difficult to address notes at specific parts.

1. Abstract: succinct and informative, but needs to be adjusted to the journal style.
2. Introduction: despite the lack of data on the role of magnesium in the dimorphism of the genus *Trichosporon*, could the authors consider including its role (and other related compounds) in fungi/yeasts? This is more important than explaining the biochemical processes involved.
3. Material and methods: I understand the complexity of the tests, but the sample size is minimal. Would it have been relevant to use at least a few more isolates to confirm the present findings? I believe it is relevant to mention it as a limitation of the study.
4. Material and methods: Please note that except in the “strains and media” subsection, references were not included. Please check and include the references used.
5. Material and methods: the used culture media, compounds, and methods were difficult to be understood. They are better described in the results (maybe because this section appears earlier according to the editorial style), but it will be worthwhile to the audience and also easier to be understood if the methods were better explained.
6. Measurement of growth rate (MM): why the authors used 72h or 144h once a single *T. asahii* isolate was used? Were the measures automatically taken every 10 minutes?
7. Figures' captions: some figures are shown incubation times different than those mentioned in the methods. Please check and correct.
8. Please, consider perform modifications in the introduction and discussion to make the manuscript easier to be comprehend.

January 20, 2023

Dear Prof. Carolina Coelho

Microbiology Spectrum

These are point-by-point responses to the Reviewer 1, Reviewer 2, and Reviewer 3. The bold texts are the comments from the Reviewers.

To Reviewer 1

Reviewer #1 (Comments for the Author):

The study by Aoki et al. describes the identification of magnesium as a limiting factor for hyphal growth in Sabouraud medium. By exploiting the presence and absence of magnesium in the growth medium, they described defects in the lipid droplets distribution and distinct mitochondrial and vacuolar morphologies in *Trichosporon asahii*. The authors described enlarged vacuoles in media supplemented with MgSO₄. The authors also proposed that yeast cells have a fragmented mitochondrial pattern, while in hypha the morphology of mitochondria was associated with filaments of actin and organized beside of the plasm membrane. In *Candida albicans*, the deprivation of magnesium inhibits the hypha formation, cell adherence and results in diminished hyphal damage during macrophage infection (PMID: 30659981; PMID: 35834579). In accordance, the absence of the micronutrient magnesium seems to be essential to the formation of hyphal forms associated with the virulence in *Trichosporon asahii*. The manuscript is technically sound and describes an interesting finding for *Trichosporon asahii* that can be valuable for future virulence studies. However, there are points that the authors might consider:

-The authors could add arrows to highlight the lipid droplets that are not associated to DNA in figure 1.

- In Fig. 1, we have highlighted the lipid droplets in arrows, that were not overlapped with DNA. The figure legend was rewritten in *line 6, page 38*.

-The figure 4A and the text do not the design of the experiment.

- The design of figure 4A was added to the legend in *line 3, page 40* as “To determine

whether JCM 2466 cells retain the ability to respond to MgSO₄ addition after continuous cell culture in Sabouraud medium, cells were cultivated for 18 h in Sabouraud medium followed by further incubation for 20 h in Sabouraud+Mg medium.” The design is written in *line 12-14, page 12*.

The mistake concerning the terms “Sabouraud/ Sabouraud+Mg” was corrected in the text (*line 14, page 12*).

-The authors suggested that the vacuoles are bigger after the addition of MgSO₄ compared with YPD. While such observation is supported by the image in figure 5, the microscopy on figure 7 shows vacuoles of similar in sizes for cells grown in YPD and Sabouraud supplemented with MgSO₄. Did the authors try somehow to quantify this phenomenon of enlarged vacuoles?

- We compared vacuolar size of JCM 2466 cells between the conditions of Sabouraud and Sabouraud+Mg media in the revised manuscript. Vacuoles were observed to shrink in fixed cells; therefore, we used live cells to examine vacuolar size in the study. We compared the size of vacuoles in cells cultured in Sabouraud and Sabouraud+Mg media by examining the area proportional to the difference in volume. For the purpose of the comparison, we measured the total vacuolar area in 86 cells, with vacuoles being classified by area for comparison in similar sized cells. Vacuoles of area <10 μm² were detected even in small cells of area <50 μm² and for cells cultured in both Sabouraud and Sabouraud+Mg media, the vacuoles increased in size with an increase in cellular area (Fig. 5C). However, whereas for the cells cultured in Sabouraud+Mg medium, the number of vacuoles of area ≥10 μm² increased as cellular area increased, this was not observed in cells cultured in Sabouraud medium (Fig. 5C). The results suggest that an increase of Mg²⁺ may induce fusion of smaller vacuoles in hyphal cells. The result (*line 17, page 13*), discussion (*line 1, page 22*), methods (*line 13, page 24*), and figure legend (*line 14, page 40*) were rewritten, respectively.

-In the passage: "Therefore, actin polymerization may be downstream in a signalling pathway...", please clarify the association made between actin polymerization signalling and the findings related to YPD and magnesium-containing medium.

- In *C. albicans*, there is a starvation-associated signaling pathway that affects expression of hyphal genes. Large lipid droplet accumulated in *T. asahii* cells, when cultivated in Sabouraud medium, indicates that nutrients are starved in Sabouraud medium. Therefore, we think that the starvation pathway functions and relates to the intake of Mg^{2+} in *T. asahii*. The term “starvation-associated signaling pathway” was added in the discussion (*line 5, page 20*).

-Is the fragmented morphology of mitochondria observed in yeast a deficiency or a pattern? Cell ghosts also had the fragmented morphology, how was evaluated the viability of the cells cultivated in Sabouraud for the microscopy?

- To examine the fragmented morphology of mitochondria in detail, we classified the distribution of mitochondria into the following three phenotypes: (1) throughout the cell cytoplasm and adjacent to the cell walls, (2) fragmented within the cytoplasm, and (3) a reduced distribution (Fig. 7C). We found that approximately 45.4% of cells grown in Sabouraud medium were characterized by mitochondria with a fragmented morphology, which was observed in both hyphal and yeast cells (Fig. 7C). In contrast, among the cells cultured in Sabouraud+Mg medium, this phenotype was mainly detected in the cells of $<10 \mu m$ (Fig. 7C). Among the cells grown in Sabouraud medium showing fragmented mitochondria, 74.6% were short cells of length $<10 \mu m$ (Fig. 7C), and among these cells, 36.4% were characterized by an unequal segregation of mitochondria and lipid droplets (Fig. 7B), which was not observed in the cells YPD and Sabouraud+Mg media. The ghost cells lacking the cytoplasm were produced by the unequal segregation (Supplemental Fig. 4) and accounted for 35.2% of the cells of length $<10 \mu m$ showing the fragmented mitochondria. Therefore, we think that the short cells showing the fragmented mitochondria are deficient; in contrast, the fragmented mitochondria in hyphal cells are a patterned phenotype. The result (*line 19, page 15*), discussion (*line 2, page 21*), and figure legend (*line 15, page 41*) were rewritten in, respectively.

We newly examined the cell viabilities grown in YPD, Sabouraud, and Sabouraud+Mg media and found that cells grown in YPD and Sabouraud+Mg media showed 100% viabilities; however, cells grown in Sabouraud medium showed 52% viability. We suppose that the lost viability of 48% was attributed to the fragmented mitochondria in the short cells of $<10 \mu m$ in Sabouraud medium. Therefore, we think that the short cells having the fragmented mitochondria is

deficient; in contrast, the fragmented mitochondria are pattern in the cells of ≥ 10 μm . The result (*line 10, page 10*), discussion (*line 4, page 21*), and the methods (*line 20, page 24*) of the colony formation assay examining viability was written, respectively.

-If morphology of mitochondria observed in yeast is different from hypha and actin is essential for hypha formation, it seems correct that the morphology of mitochondria would be the same as observed in yeast with the treatment of latrunculin A. Would the mitochondria morphology change if, after hypha formation, it was added latrunculin A and incubated for 4-5 hours?

- We already showed that actin was important for hyphal formation because cell length apparently shortened, and a swollen-shaped phenotype was produced when latrunculin A was added into each media (Fig. 6). When latrunculin A was added, the mitochondrial distribution also changed to dot-like distributions (Supplemental Fig. 5). However, the change of mitochondrial distribution might be caused by the change of cell length as Reviewer 1 pointed out. To show that actin is essential for mitochondrial distribution, we newly examined the morphology of the mitochondria 5 h after treating the cells with either latrunculin A or DMSO to cells cultivated for 16 h in YPD, Sabouraud, and Sabouraud+Mg media. We found that the mitochondrial distribution was changed into a dot-like pattern within the cytoplasm by adding latrunculin A, even in hyphal cells cultivated in YPD and Sabouraud+Mg media (Fig. 9A and C). Therefore, we think that the mitochondrial distribution in *T. asahii* is associated with the actin cytoskeletal network in *T. asahii*. The result (*line 3, page 17*), discussion (*line 23, page 20*), and figure legend (*line 13, page 42*) were written, respectively.

-In the passage: "The length of *T. asahii* cells gradually decreased..", are the cells diminishing or the higher concentrations of magnesium giving better support for hyphal growth?

- We wrote the sentence to show that the average length of the cells shortened as the concentration of MgSO_4 was decreased in the medium. The term "decreased" was corrected to "shortened" in the text (*line 13, page 20*).

-I would suggest that the authors revise their conclusions about the role of Mg²⁺, transporters and gene expression, as some of them sound very speculative and are not supported by the findings described in the manuscript.

- Previous reports showed that the changes associated with Mg²⁺ and magnesium transporters was allosteric; however, hyphal growth induced by adding MgSO₄ was not allosteric in Fig. 2D. Therefore, we speculate that the morphological changes in the study are caused by the action of Mg²⁺ inside the cell. In *T. asahii*, the magnesium transporter genes are not reported yet; therefore, we have deleted the gene names and revised the sentence in the discussion (*line 14, page 20*) to be less definitive. The discussion about gene expression was deleted.

To Reviewer 2

Reviewer #2 (Comments for the Author):

Hyphal growth in *Trichosporon asahii* is accelerated by the addition of magnesium. Aoki et al. This paper seeks to examine factors contributing to hyphal formation of *Trichosporon asahii* as well as intracellular organisation in defined and rich media. The authors show that the addition of magnesium accelerates hyphal growth for a limited time. Distribution and morphology of lipid droplets, vacuoles, cortical actin patches, and mitochondria were examined in cells cultured in different growth media.

However, authors compare organelle distribution in yeast cells with polarised hyphal forms, attributing difference in intracellular organisation to media composition, while failing to compare the same morphological forms. Microscopy images are not sufficiently clear to serve as evidence behind the conclusions drawn.

Major comments:

1. Cell length measurements seem to be highly variable, with SD exceeding mean for many conditions. Since the authors aimed to examine hyphal induction, there could be more suitable ways to present the data - for example, in a form of a morphology index.

- We showed the data of cell length measurements in the form of morphology index in Fig. 2A, 2B, 2C, 4B, and 6A. To form the morphology index, the number of cells were classified into eleven groups of cell length ranged as follows: 0–10 μm , 10–20 μm , 20–30 μm , 30–40 μm , 40–50 μm , 50–60 μm , 60–70 μm , 70–80 μm , 80–90 μm , 90–100 μm , and longer than 100 μm . The result was rewritten in the text (*line 12, page 8; line 18, page 8; line 4, page 9; line 8, page 9; line 15, page 9; line 17, page 12; line 16, page 14*).

2. Unclear microscopy images mainly showing medial cell plane in fixed cells, while drawing conclusions on organelle co-localisation and their dynamics in cells grown in different media. For example, in Fig 7, plasma membrane staining is required to demonstrate mitochondrial distribution adjacent to the membrane. Fig 8 attempts to show association of mitochondria with actin patches, but images are not clear and no control to account for background fluorescence is shown.

- To show the medial cell plane in Fig. 7, we used Calcofluor White reagent that

stained the cell wall of fungi. A proportion of the mitochondria was observed to be distributed adjacent to the cell walls, detected using Calcofluor White, in YPD and Sabouraud+Mg media (Fig. 7A and B). In addition, a proportion of mitochondria was also distributed throughout the cell cytoplasm in hyphal and yeast-like cell in YPD and Sabouraud+Mg media. Therefore, we revised the mitochondrial distribution as throughout the cell cytoplasm and adjacent to the cell walls. The distribution of mitochondria adjacent to the cell walls is still a marker showing the mitochondrial distribution in the cells supplemented with MgSO₄. The cell images of hyphal cells and yeast-like cells were described in Fig. 7A and B, respectively. The result (*line 10, page 15*), methods (*line 21, page 25*), and figure legend (*line 10, page 41*) were rewritten, respectively.

Concerning co-staining of actin and mitochondria, we deleted all the data because paraformaldehyde-fixed *T. asahii* cells produced background signals with the green filter.

3. Differences in organelle morphology and distribution could largely be explained by the lack of polarised growth and the cell cycle arrest in cells cultured in Sabouraud medium. For example in Fig 5, cells in the same growth stages need to be examined for vacuolar morphology.

- We appreciate the encouragement. *T. asahii* cells are classified into three morphological types: yeast, hypha, and arthroconidia, and the size of each type is very variable. In the study, it is difficult for us to describe the relationships about polarized growth and cell cycle arrest that Reviewer 2 pointed out. We would like to understand the association between organelle morphology and polarized growth or cell cycle in a future study. Here, we analyzed the vacuolar size and frequency of mitochondrial distribution by ranging each cellular size in Fig. 5C and 7C, respectively. To quantify the size of vacuoles, we compared the size of vacuoles in cells cultured in Sabouraud and Sabouraud+Mg media by examining the area proportional to the difference in volume. For the purpose of the comparison, we measured the total vacuolar area in 86 cells, with vacuoles being classified by area for comparison in similar sized cells. Vacuoles of area <10 μm^2 were detected even in small cells of area <50 μm^2 and for cells cultured in both Sabouraud and Sabouraud+Mg media, the vacuoles increased in size with an increase in cellular area (Fig. 5C). However, whereas for the cells cultured in Sabouraud+Mg medium,

the number of vacuoles of area $\geq 10 \mu\text{m}^2$ increased as cellular area increased, this was not observed in cells cultured in Sabouraud medium (Fig. 5C). The results suggest that an increase of Mg^{2+} may induce fusion of smaller vacuoles in hyphal cells. The result (*line 17, page 13*), discussion (*line 1, page 22*), methods (*line 13, page 24*), and figure legend (*line 14, page 40*) were written, respectively.

To examine the fragmented morphology of mitochondria in detail along cell length, we classified the distribution of mitochondria into the following three phenotypes: (1) throughout the cell cytoplasm and adjacent to the cell walls, (2) fragmented within the cytoplasm, and (3) a reduced distribution (Fig. 7C). We found that approximately 45.4% of cells grown in Sabouraud medium were characterized by mitochondria with a fragmented morphology, which was observed in both hyphal and yeast cells (Fig. 7C). Therefore, appearance of the fragmented mitochondria may be independent of the difference in cell length. The result (*line 19, page 15*), discussion (*line 2, page 21*), and figure legend (*line 15, page 41*) were written, respectively.

4. Conclusions drawn in Results and Discussion sections regarding organelle co-localisation, mitochondrial activity, functional lipid droplets, and actin polymerisation have not been convincingly substantiated in this paper.

- Mitochondrial distribution adjacent to the cell walls was stained with Calcofluor White reagent (Fig. 7A). Immuno-fluorescence data of actin was deleted because paraformaldehyde-fixed *T. asahii* cells had background signals with the green filter. The direct association of actin and mitochondria was not elucidated in the study. However, the mitochondrial distribution was inhibited in hyphal cells by the actin inhibitor (Fig. 9; Supplemental Figure 4), suggesting a potential interaction. The discussions about the morphological changes of organelles in *T. asahii* was revised based on previous reports and data in the study and was rewritten in the discussion to be less definitive (*line 23, page 20; line 11, page 21; line 1, page 22; line 10, page 22*).

We did not estimate whether lipid droplets were functional or not; however, the accumulation of large lipid droplet in Sabouraud medium (Fig. 5A) indicated that lipid may not disperse throughout the cells cultivated in Sabouraud medium. Therefore, it is suggested that nutrients may not be supplied normally to the mitochondria in the cells cultivated in Sabouraud medium in *line 21-24, page 21*.

Conclusion about mitochondrial activity was deleted from the discussion.

Minor comments:

5. Page 7, paragraph 2. Clarify whether OD600 or OD660 was used.

- In *line 17, page 7*, OD600 was corrected to OD660.

6. Fig 2B. Statistical analysis should compare salt drop-out media with media + salt to determine which ions were required for growth, and not with MgSO₄Δ.

- Each statistical analysis of the drop-out media with media + salt was added in the Supplemental Table. Statistical data are described only in the Supplemental Table.

7. Fig 2D. Actual concentrations should be included on the figure and in the supplemental table.

- Each concentration of MgSO₄ was written in *line 7, page 39* in the figure legend of 2D and in the Supplemental Table.

8. Fig 3C. Hyphal width looks significantly different in Sabouraud medium and Sabouraud with MgSO₄. This should be acknowledged in the results.

- Thank you for the suggestion of morphology. We measured the cell width in Sabouraud and Sabouraud+Mg media. The cell width in Sabouraud medium gradually increased from 3.02 μm at 0 h to 3.81 μm at 144 h. In contrast, the cell width in Sabouraud+Mg medium increased from 3.02 μm at 0 h to 3.57 μm at 8 h, but subsequently declined to 2.31 μm when measured at 48 h (Fig. 3C; Supplemental Table). From 48 h to 144 h, the cell width gradually increased to 2.63 μm ; however, the cell width was shorter in every time points in Sabouraud+Mg medium than in Sabouraud medium (Fig. 3C). The width of the JCM 2466 cells cultivated in Sabouraud medium was 1.45-fold wider than that of those cultivated in Sabouraud+Mg medium (Fig. 3B; Supplemental Table). An increase of Mg²⁺ may

restrict cell width. A graph was provided in Fig. 3C. The result (*line 21, page 11*), discussion (*line 6, page 19*), and figure legend (*line 20, page 39*) were written.

9. Page 11, paragraph 2. Fig 4 shows that magnesium can accelerate hyphal germination and elongation of the cells pre-incubated in Sabouraud medium. Claiming that these data show that cells retain the ability to form hyphae after 18h in Sabouraud medium is incorrect, when Fig 3 clearly shows hyphae are able to form in Sab alone.

- We corrected the sentence to “JCM 2466 cells could still respond to the addition of MgSO₄” (*line 12, page 12; line 19, page 12*).

10. Page 12 line 1. Dynamics of lipid droplets - "morphology" might be better suited for describing these differences.

- The term “dynamics of lipid droplets” was corrected to “morphology of lipid droplets” (*line 6, page 13*).

11. Use of language: vitamin mix or vitamins - not vitamin in a singular form to describe combination of compounds.

- The singular form of vitamin was corrected to “vitamins” in the text (*line 6, page 8*).

12. Fig 7B: a cross section of the hyphal form should also be shown for cells grown in Sabouraud medium alone.

- A part of the hyphal form was detected in Sabouraud medium in Fig. 8B. The hyphal form was indicated by an arrow. A medial plane of the hyphal cell was indicated by the dotted lines. A black arrowhead indicated a horizontal orange line that showed the position of a cross section of the hyphal form for observing the side view. The side view of the hyphal form was attached under the merged image of Sabouraud medium. The figure legend was rewritten in *line 6, page 42*.

13. P17 line 18-22. These conclusions are confusing - consider revising.

- We observed that the intake of Mg^{2+} induced hyphal growth, and that cells were swollen-shaped upon the addition of an actin inhibitor. From these observations, we first thought that actin polymerization pathway was downstream of Mg^{2+} signaling. In contrast, another possibility was also considered. Mg^{2+} uptake may increase the cell volume, but not the length of *T. asahii* cells, because the cell volume of cells grown in Sabouraud+Mg medium was larger than that of those in Sabouraud medium, under the inhibition of actin (Fig. 6B; Supplemental Fig. 4). In this case, a signaling pathway affected by Mg^{2+} may be independent of the actin polymerization. The possibility was rewritten in the discussion (*line 6, page 20*). We would like to clarify the mechanism by which an increase of Mg^{2+} affects hyphal growth in a future study.

To Reviewer 3

Reviewer #3 (Comments for the Author):

The manuscript presents data on the mechanism of yeast-to-hypha transition of *Trichosporon asahii* and identifies the factors that induce its hyphal growth. The data are original and there are no reports in the literature about the factors that stimulate the dimorphism of *T. asahii*. The results found are interesting and will contribute significantly to the target scientific community. The manuscript is well written and organized.

Here are some considerations.

1. Format the list of references according to the style of the journal.

- The list of references was formatted according to the style of Microbiology Spectrum.

2. Figure 3: Insert the image of cells at time 0 in Sabouraud + MgSO₄ medium

- In Fig. 3D, an image at time 0 was added in the line of Sabouraud + Mg medium.

February 21, 2023

Dr. Keita Aoki
Tokyo Nogyo Daigaku
Tokyo NODAI Research Institute
1-1-1 Sakuragaoka
Setagaya
Tokyo, Tokyo 156-8502
Japan

Re: Spectrum04242-22R1 (Hyphal growth in *Trichosporon asahii* is accelerated by the addition of magnesium)

Dear Dr. Keita Aoki:

The manuscript is improved from previous version. There are still some issues to resolve: Reviewer #1 raises issues of cell viability in new measurements, which should be relatively easy to perform.

Link Not Available

Sincerely,

Carolina Coelho

Journals Department
Reviewer comments:

Reviewer #2 (Comments for the Author):

The authors have addressed the comments and included additional experiments where reasonably possible. Importantly, cells of similar lengths were compared for organelle morphology and distribution, and additional data quantification was included. However, authors report a new finding of a significant cell viability decrease in Sabouraud medium compared to YPD and Sab+Mg, which might contribute to the aberrant morphologies observed.

Additional points:

Percentage of cells corresponding to 11 length groups is informative and adds to the interpretation of the results. However, an index normally refers to a ratio of several measurements (e.g. a morphology index in *C. albicans* is a mathematical ratio between maximum cell length, maximum diameter and septal diameter; Merson-Davies & Odds. J Gen Microbiol. 1989). Therefore, terminology such as cell length is sufficient to describe these results.

Distribution of mitochondria does appear different in hyphae cultured in Sab alone (Fig7A). However, since the authors reported 48% cell death in Sab medium, can the authors demonstrate that these cells were viable before staining? The same question of cell viability would apply to vacuole size measurements and lipid droplet morphology.

Staff Comments:

Preparing Revision Guidelines

Please return the manuscript within 60 days; if you cannot complete the modification within this time period, please contact me. If you do not wish to modify the manuscript and prefer to submit it to another journal, please notify me of your decision immediately so that the manuscript may be formally withdrawn from consideration by Microbiology Spectrum.

March 23, 2023

Dear Prof. Carolina Coelho

Microbiology Spectrum

We would like to thank reviewer 2 for the constructive critique to improve the manuscript. We have made every effort to address the issues raised and responded to all comments. Please, find next a detailed, point-by-point response to the reviewer's comments. We hope that our revisions will meet the reviewer's expectation.

To Reviewer 2

Reviewer #2 (Comments for the Author):

The authors have addressed the comments and included additional experiments where reasonably possible. Importantly, cells of similar lengths were compared for organelle morphology and distribution, and additional data quantification was included. However, authors report a new finding of a significant cell viability decrease in Sabouraud medium compared to YPD and Sb+Mg, which might contribute to the aberrant morphologies observed.

- To separate viable cells from dead cells under the microscope, we stained the cells using SYTOX Green, which penetrates only damaged cell membranes. Furthermore, the reduction of viability of the cells cultured in Sabouraud medium was confirmed by the staining experiment and was comparable to that measured using the colony formation assay (*line 14, page 10*) (Supplemental Figure 4A).

In the experiment, both yeast-like cells and hyphal cells were stained using SYTOX Green. The cells possessing large lipid droplets or fragmented mitochondria were not necessarily stained by SYTOX Green. The result corresponded to the presumption that fragmented mitochondrial distribution is a patterned distribution and not a defective distribution in the discussion (*line 3, page 22*). When the organelles were unequally segregated, only 60.5% of the cells including fragmented mitochondria and large lipid droplets were stained by SYTOX Green (*line 11, page 16*). Therefore, we speculate that the mis-segregation of fragmented mitochondria and large lipid droplets may cause the reduction of viability in Sabouraud medium (Supplemental Figure 4D). In any case, because the reduction of viability would be affected not only by aberrant morphologies but also

by a decrease of cellular metabolism, we need to study the mechanism in more detail in the future.

Additional points:

Percentage of cells corresponding to 11 length groups is informative and adds to the interpretation of the results. However, an index normally refers to a ratio of several measurements (e.g. a morphology index in *C. albicans* is a mathematical ratio between maximum cell length, maximum diameter and septal diameter; Merson-Davies & Odds. *J Gen Microbiol.* 1989). Therefore, terminology such as cell length is sufficient to describe these results.

- “Morphology index” was changed to “cell length classification” in the text.

Distribution of mitochondria does appear different in hyphae cultured in Sab alone (Fig7A). However, since the authors reported 48% cell death in Sab medium, can the authors demonstrate that these cells viability would apply to vacuole size and lipid droplet morphology.

- We stained the cells using SYTOX Green and succeeded in separating viable cells and dead cells under the microscope. In Supplemental Figure 4D, a hyphal cell, which possessed the fragmented mitochondria and large lipid droplets but was not stained with SYTOX Green, was described. A yeast cell stained with SYTOX Green was also shown. In both cell types, a cell compartment that included large lipid droplets was rarely stained with SYTOX Green. An unknown mechanism that produces large lipid droplets might exist and, therefore, should be studied in the future.

Furthermore, the experiment using SYTOX Green uncovered a difference in the vacuolar area between viable cells and dead cells. The total vacuolar area per cell was approximately three times larger in SYTOX-negative (viable) cells than in SYTOX-positive (dead) cells (*line 7, page 14*) (Supplemental Figure 4B).

April 4, 2023

Dr. Keita Aoki
Tokyo Nogyo Daigaku
Tokyo NODAI Research Institute
1-1-1 Sakuragaoka
Setagaya
Tokyo, Tokyo 156-8502
Japan

Re: Spectrum04242-22R2 (Hyphal growth in *Trichosporon asahii* is accelerated by the addition of magnesium)

Dear Dr. Keita Aoki:

The authors improved the manuscript according to reviewer's comments.

Your manuscript has been accepted, and I am forwarding it to the ASM Journals Department for publication. You will be notified when your proofs are ready to be viewed.

Sincerely,

Carolina Coelho
Editor, Microbiology Spectrum
